# CHAINING SPECTRAL PEARLS: ELLIPSOIDAL FORE-CASTING BEYOND TRAJECTORIES FOR TIME SERIES

## ABSTRACT

Current long-term time-series forecasting (LTSF) benchmarks are dominated by noisy stochastic datasets and pointwise losses, so models that look strong on ETT-type tasks can behave unpredictably under deterministic chaos or controlled regime shifts. We argue that forecasters should be stress-tested on canonical chaotic systems and on synthetic benchmarks with precisely scripted non-stationarity, and that evaluation should focus on the geometry of predictive distributions, not just single trajectories. We present FERN (Forecasting with Ellipsoidal RepresentatioN), a geometry-aware forecaster that uses a bidirectional encoder and a per-patch local linear transport map, factored as translate–rotate–scale–rotate-back with explicit eigenvalues and eigenvectors. The network therefore "only" learns to generate stable Jacobians, while users obtain spectral diagnostics of local stretching, volume change, and regime switches. Alongside MSE/MAE we report Wasserstein Distance (shape fidelity) and Effective Prediction Time (horizon stability). Across 21 synthetic systems (chaotic, stochastic, and switching) and cleaned ETT/Weather benchmarks, FERN is a strong all-round "safe" model: it achieves the best or second-best MSE or SWD on 19/21 synthetic tasks, maintains geometric fidelity far beyond the Lyapunov horizon on Lorenz-63, and remains competitive on real-world LTSF. The codebase also releases our controlled-shock benchmark and data-cleaning protocol.

## 1 INTRODUCTION

Time-delay embedding (TDE) represents a dynamical system's state as a vector of consecutive observations, e.g., $[x_t, x_{t-1}, \ldots, x_{t-k}]$. By Takens' theorem Takens (1981) and its stochastic extensions (Stark et al., 1999; Stark, 2003), a sufficiently long TDE of a single observable can reconstruct the system's attractor (set of states the system evolves toward) or invariant measure up to a smooth change of coordinates. Intuitively, it means each channel's historical context encodes system-wide qualitative information. **Long-term time series forecasting (LTSF)** is then the task of predicting future TDE $[y_1, \ldots, y_n]$ from a past TDE $[x_1, \ldots, x_n]$, typically with $y_1 = x_{n+1}$ across channels.

The unpredictability in time-series data comes from three entangled sources: *stochastic noise, deterministic chaos, and non-stationary regime shifts*. Forecasting is hard because real-world data mix all three, yet models that excel at one often struggle with the others. General-purpose forecasting demands robustness to chaos and noise, and adaptability to regime shifts.

Prediction alone is insufficient. *Long-horizon forecasts will inevitably fail*, the value of a model depends on how much *structural* insight it provides to *decision making*: does it quantify uncertainty, provide geometric insight into why a prediction looks the *way* it does, and offer **actionable interpretability**?

We seek a *unified framework* that handle jointly three entangled sources of unpredictability: chaos, stochasticity, and non-stationarity. We believe modern forecasting tools should provide explicit spectral structure—eigenvalues and eigenvectors—that completely characterizes the learned linear dynamics. These are not merely interpretable; they unlock decades of mature tools for stability analysis, uncertainty quantification, and intervention.

Yet spectral structure is of limited value if it comes from a brittle model. Hand-crafted priors—trend-seasonal decompositions, fixed Fourier bases—fail under regime shifts. Following Sutton's

Bitter Lesson Sutton (2019), we want fully data-driven learning that scales with computation, while also exposing interpretable structure.

This creates a conflict: to model the **true time-varying dynamics** $F_t(\cdot)$, we need the **flexibility** of a general neural approximator $\hat{F}_t(\cdot)$, but also the **structure** of a linear Jacobian to access spectral properties. Direct Jacobian modeling is computationally prohibitive: (1) there are $O(n^2)$ Jacobian entries for an $n$-dimensional prediction horizon; (2) this matrix has $O(n^3)$ eigen-decomposition cost; (3) when n is too large, naive approaches require either prohibitive global cost or autoregressive patch-by-patch generation, sacrificing parallelism.

Our main contributions are as follows:

1. **Local transport paradigm.** We address these three computational bottlenecks by shifting from integrating global vector fields to learning **local transport maps**. This allows us to approximate the conditional forecast distribution using computationally efficient, parallelizable operators rooted in optimal transport (OT).

2. **The FERN Model.** We implement this paradigm via patch-wise linear transforms with positive semi-definite (SPSD) Jacobians. This parameterization directly outputs interpretable eigenvalues (scaling) and eigenvectors (rotation) without expensive eigen-decomposition; the outputs are intuitive Gaussian-ellipsoids.

3. **Physically interpretable benchmarks.** We introduce a synthetic suite that isolates chaos, stochasticity, and non-stationarity, including three types of controlled regime-shift experiments whose timing, magnitude, and before/after distributions are fully known.

4. **Evaluation protocol.** We complement MSE/MAE with geometry-aware distributional metrics (Wasserstein-2 / sliced Wasserstein) and an effective prediction time (EPT) metric that quantifies the horizon of reliable pointwise prediction. We identify data-quality issues and recency bias in standard LTSF tasks, and propose a transparent preprocessing and validation scheme.

## 2 METHODS

### 2.1 NEW FORECASTING APPROACH

For LTSF, the target at time $t$ is the *empirical distribution* induced by the true dynamics $F_t(\cdot)$. **Wasserstein-2 distance (W2)** (Peyré & Cuturi, 2019) measures the minimal "work" (expected squared Euclidean cost) required to transport one distribution into another.

**Brenier's theorem Brenier (1991) addresses three computational bottlenecks simultaneously**: for source distribution $\nu$ and target distribution $\eta$ (empirical measure induced by dynamics $F_t(\cdot)$), under regularity conditions ($\nu$ is absolutely continuous and $\eta$ has finite second moments), there *exists* a *unique*, Wasserstein-2 **optimal map** $G_{\nu,\eta}(\cdot)$ from $\nu$ to $\eta$, which can be written as the *gradient of a convex potential* and therefore has a **symmetric positive semi-definite (SPSD)** Jacobian almost everywhere. Crucially, while the true dynamics $F_t(\cdot)$ may be arbitrary and hard to learn, Brenier guarantees that a structure-preserving map $G_{\nu,\eta}(\cdot)$ reaching the exact same target **always exists**. From the *search* perspective, this justifies constraining the hypothesis space in MSE-only training: we trade the ill-posed problem of "finding zero-MSE solution among all functions and hope that approximate $F(\cdot)$ " for the *easier* problem of "finding zero-MSE solution within the convex cone of SPSD maps, and hope that approximate $G_{\nu,\eta}(\cdot)$". Since any SPSD matrix admits the decomposition $U\Lambda U\top$ where $U$ is orthogonal (eigenvectors) and $\Lambda$ is diagonal with nonnegative entries (eigenvalues), we can *parameterize the Jacobian through its spectral factors directly*.

**How to parameterize:** $\Lambda$ is an $n$-dimension vector of scales which costs $O(n)$ to generate with an MLP; any orthogonal matrix $U$ decomposes into at most $n$ **Householder reflections** $x \mapsto (I - 2vv^\top)x$ where $v$ are MLP-generated, unit-norm reflection vectors. Each reflection costs $O(n)$. Let $R$ denote the number of reflections. Setting $R = n$, the full orthogonal group is reachable, meaning the search space is sufficiently expressive to recover any target rotation. Crucially, setting $R < n$ enables $O(R \cdot n)$ cost *reduced-complexity approximation*. The overall Jacobian search cost under this SPSD parameterization is reduced from $O(n^2)$ to $O(R \cdot n)$, and *eigenvalues and eigenvectors are directly available without incurring the $O(n^3)$ cost*.

**Patching *further* accelerates computation.** Let $p$ be the patch size, $n_p$ be the number of patches and $n_p = n/p$. Applying Brenier's theorem to patches yields $n_p$ Jacobians of the form $U\Lambda U^\top$, each with $O(p^2)$ (total $O(n_p \cdot p^2) = O(n \cdot p)$) and $O(R \cdot p)$ (total $O(n_p \cdot R \cdot p) = O(R \cdot n)$) cost for full and reduced-complexity search respectively. Importantly, since each patch's movement does *not* depend on the prediction from the previous patch, we can **make predictions in parallel**. For the Lorenz-63 Lorenz (1963) system with prediction horizon 336, 10 delay dimensions is a very conservative setting to invoke Takens theorem. If we take 14 size-24 patches and set R=8, the cost is reduced from $O(336^2)$ to $O(24 \cdot 336)$; a reduced-complexity version costs $O(8 \cdot 336)$.

**In summary,** by invoking Brenier's theorem, we turn an autoregressive and costly general search problem into a structured, parallelizable, and efficient search. Fig. 1 demonstrates actionable interpretability of our prediction FERN on Lorenz-63. In the ground truth (left), the Lorenz-63 system exhibits distinct kinematic phases: it accelerates along the outer rings (deep blue) and decelerates significantly at the "bottleneck" of the transition zone (almost white). Remarkably, FERN's maximum eigenvalue is highest precisely in the high speed regions (dark orange, right) and lowest in the slow speed regions, reflecting the model's *learned belief* that large stretching is needed where the system moves fast.

Crucially, since these eigenvalues act on a fixed Gaussian source, they directly encode *what the model believes during prediction* about local instability, independent of the ground-truth speed. They act as a learned (without supervision), location-dependent confidence signal, and they are *comparable* across patches and forecast runs. The structure is the diagnostic.

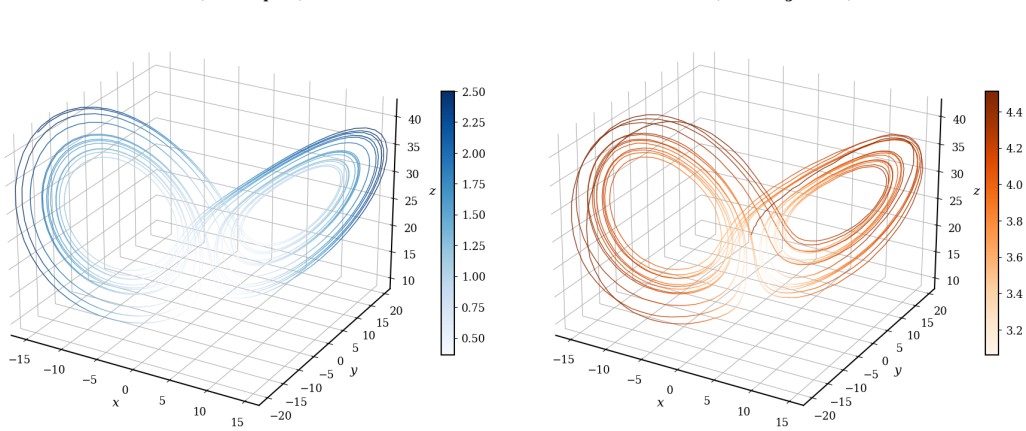

Figure 1: Lorenz-63 attractor reconstruction with FERN. Left: ground-truth trajectory colored by instantaneous speed (norm of velocity). Right: FERN prediction, colored by the mean patch-wise maximum eigenvalue (spectral radius of the local SPSD map).

**Local Geometry: Gaussian Ellipsoids.** The unstructured, general form of $\hat{F}(\cdot)$ is replaced by the SPSD Jacobian of $G(\cdot)$, one of the most structured and intensively studied matrices in linear algebra and statistics. Let $A := U\Lambda U^\top$ be the data-dependent Jacobian generated for a patch by the network, and let $t$ be a data-dependent translation with $O(n)$ cost. Applying $A$ and $t$ to Gaussian noise always yields a *Gaussian ellipsoid* $N(t, AA^\top) = N(t, U\Lambda^2 U^\top)$, which we call the **local geometry** of that patch.

The map $y \mapsto U\Lambda U^\top y + t$ is a simple geometric sequence: rotate, scale anisotropically, rotate back, then translate. Translation does not affect the Jacobian, but it plays the role of a residual correction to a first-order Taylor expansion:

$$T(y) \approx T(y_0) + J_T(y_0)(y - y_0) = U\Lambda U^\top y + T(y_0) - U\Lambda U^\top y_0, \tag{1}$$

where $t := T(y_0) - U\Lambda U^\top y_0$. In this decomposition, the piecewise Jacobians $A$ determine the local Gaussian ellipsoid shape ($U$ and $\Lambda$ control orientation and spread), while the translations $t$ adjust the

---

**Algorithm 1** FERN

---

**Require:** Windowed input $x$ with shape [Batch, Feature, Sequence_length]

1: $z \leftarrow \mathcal{N}(0, I); \quad y_0 \leftarrow \mathcal{N}(0, I)$
2: $x^1 \leftarrow x, z^1 \leftarrow z$
3: **for** $i = 1$ **to** $K_{\text{enc}} = 5$ **do**
4: $\quad z^{i+1} \leftarrow \mathbf{s}(x^i) \odot z^i + \mathbf{t}(x^i)$
5: $\quad x^{i+1} \leftarrow \mathbf{s}(z^i) \odot x^i + \mathbf{t}(z^i)$ {*Coupling $x \leftrightarrow z$*}
6: **end for**
7: $y_\mu^1 \leftarrow y_0$
8: **for** $j = 1$ **to** $K_{\text{dec}} = 3$ **do**
9: $\quad y_\mu^{j+1} \leftarrow y_\mu^j + \mathbf{t}(z)$
10: $\quad z^{j+1} \leftarrow \mathbf{s}(y_\mu^j) \odot z^j + \mathbf{t}(y_\mu^j)$ {*Coupling $z \leftrightarrow y_\mu$*}
11: **end for**
12: $y_{\text{rot}} \leftarrow U(z)\, y_\mu$ {rotate}
13: $y_{\text{scaled}} \leftarrow \Lambda\, y_{\text{rot}}$ {nonnegative anisotropic scaling}
14: $y_{\text{unrot}} \leftarrow U(z)^\top y_{\text{scaled}}$ {rotate back}
15: $y^* \leftarrow y_{\text{unrot}}$ {prediction in data space}
16: **return** $y^*$

---

mean. Geometrically, this separates *how* the model stretches noise—the model's belief about local uncertainty, from *where* it places the forecast——absorbing errors from the linear approximation.

**Bidirectional coupling network encoding.** It remains to specify how the data-dependent spectral factors $\Lambda(z)$, $U(z)$, and $t(z)$ are generated. We encode the input window $x$ into a Gaussian latent $z$ via a bidirectional coupling network: $x$ and $z \sim \mathcal{N}(0, I)$ iteratively refine each other through learned scale-and-shift operations, with $z$ remaining Gaussian throughout. A second coupling stage refines $y_0 \sim \mathcal{N}(0, I)$ using the encoded $z$, but only updates $y_0$ via shifts (mean refinement) before the final SPSD linear transform.

Fig. 2 shows a schematic illustration of FERN. Architecturally, three large MLPs ($H_x, H_y, H_z$; each 4 linear layers) map their respective inputs to a shared *hidden space* $h$ (we reserve the term *latent space* for $z$). Smaller head networks (1–2 layers) read from $h$ to produce the scales and shifts applied to coupling partners. At inference, a fresh $z$ is drawn each forward pass; the MLPs then output $U(z)\Lambda(z)U(z)^\top + t(z)$. See Algo. 1 for entire details. This structure is inspired by Augmented Normalizing Flows (ANF) (Huang et al., 2020), adapted here for conditional prediction rather than density estimation.

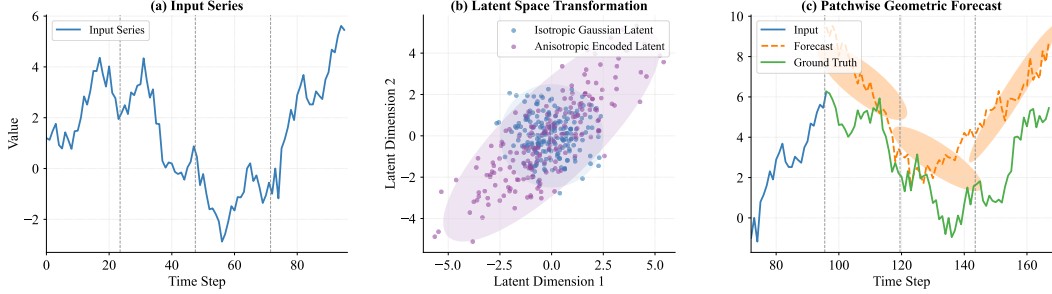

Figure 2: FERN forecasting mechanism. (a) Input time series to be processed by the bidirectional encoder. (b) Latent noise $z \sim \mathcal{N}(0, I)$ is encoded as Gaussian ellipsoids. (c) Output space noise $y_0 \sim \mathcal{N}(0, I)$ is transformed into prediction (patches of Gaussian ellipsoids).

## 3  NEW EVALUATION PROTOCOL

### 3.1  DISTRIBUTIONAL METRICS

**What pointwise metrics miss.**   Pointwise metrics critically penalize phase shifts. For instance, $[0, 1, 2, 3, 4]$ and $[1, 2, 3, 4, 5]$ are intuitively similar, as are $[0, 2, 4, 6, 8]$ and $[2, 4, 6, 8, 10]$, yet the same one-step shift yields MSEs of 1 and 4, heavily penalizing phase. Pointwise metrics leave *no room for error*: correct predictions that arrive slightly too early or too late are treated as entirely wrong, whereas human judgment recognizes shape similarity and phase shifts.

**Wasserstein-2 Distance (W2).**   While DTW (Sakoe & Chiba, 1978) partially addresses phase, it is computationally heavy and non-differentiable. The **Wasserstein-2 (W2)** distance introduced earlier provides a rigorous geometric solution. W2 measures the minimal "work" required to morph the predicted distribution into the ground truth. Unlike local divergences (KL, Hellinger, MMD) that favour averaged predictions, W2 rewards sharp forecasts even under phase shift.

In LTSF, each channel is a 1D time-delay embedding, and W2 simplifies elegantly to the $L_2$ distance between *sorted* values (quantiles) Peyré & Cuturi (2019):

$$W_2^2(\mu^\star, \mu) = \int_0^1 \left( F_{\mu^\star}^{-1}(u) - F_\mu^{-1}(u) \right)^2 du = \frac{1}{H} \sum_{h=1}^{H} \left( y_{(h)}^* - y_{(h)} \right)^2,$$

where $y_{(1)}^* \leq \cdots \leq y_{(H)}^*$ and $y_{(1)} \leq \cdots \leq y_{(H)}$ are the *order statistics*. This makes W2 an **index-agnostic** shape metric: it compares the histogram of the forecast to the histogram of the truth, completely ignoring temporal alignment. It complements MSE (which enforces index-alignment) by quantifying: *"Did the model predict the correct set of values, regardless of exactly when they occurred?"*. We did use both 1D-WD and its generalization SWD, see Appendix A.5 for distinction and Appendix A.7 for setup.

**Effective prediction time (EPT).**   We complement W2 with Effective Prediction Time (EPT), defined as the first forecast step at which the error exceeds one standard deviation of the training data. EPT quantifies reliability: if multiple models have EPT $\approx 190$ on a 336-step horizon, steps $t > 190$ should be treated more as failure-mode analysis than actionable signal.

### 3.2  SYSTEMATIC STRESS TESTING WITH SYNTHETIC DATASETS

**Real-world datasets**. These are *single realizations* of an **unknown data-generating process, with an unknown intervention history**, complicated by noise, nonstationary shocks, data quality issues. While important, overemphasis risks rewarding **historical path emulation**, i.e., matching the particular realized path among many counterfactuals.

**Synthetic datasets.** Their underlying data-generating processes are *known*. They are easy to visualize, and lightweight to simulate—an essential complement to real-world datasets. In particular, *chaotic datasets* are an empirically important yet under-tested class in LTSF. **Lyapunov time** specifies the time for the distance between nearby trajectories to increase by a factor of e. A defining property for chaotic systems: their Lyapunov times are short. Known as *sensitive dependence on initial conditions*, any infinitesimally small error grows exponentially. Long-term *pointwise predictions* are impossible, even in theory; they quickly become uninformative beyond a few Lyapunov times.

**Why stress testing on chaotic datasets?**   (1) Short-term chaotic behaviour is widespread in real-world (even in ETT datasets Zhou et al. (2021), see Hu et al. (2024)). Models untested on this class of challenging deterministic class can be fatally brittle to short term chaos with error explosions. As forecast horizon increases, especially beyond *pointwise* predictability, we observe long term behaviour: does the model gracefully settles on an average prediction, or fluctuates wildly to mimic chaos?

**Why stress testing against *explicit* non-stationarity?**   Better MSE on ETT dataset *may or may not* mean that the implicit non-stationarity has been handled. Worse, when the validation and test

errors diverge, we cannot distinguish overfitting from regime shift as we don't know its timing. We prefer: non-stationarity with its type, timing, magnitude, and before/after distributions all explicit and precise, and no unknown shocks happen between validation and test, unless explicitly designed. We design three controlled non-stationarity scenarios, **all occurring exactly at the midpoint of the training trajectory** (Full numerical settings in Appendix A.7.1):

- **Parameter drift:** system parameters shift slightly (same initial conditions).
- **State perturbation:** state variables receive an additive shock (same parameters).
- **Regime replacement:** the trajectory switches to a different one (different parameters *and* initial conditions).

### 3.3 DATA QUALITY AND CHECKPOINT STRATEGY

**Data Preprocessing.** Real-world benchmarks contain artifacts absent from synthetic data. In ETTh2, two columns have 22–33% zeros occurring in multi-day consecutive runs; sentinel values like 88.297 persist for months 4 (Fig. 4). Standard MSE/MAE losses assume continuous Gaussian/Laplacian errors and are ill-equipped for such zero-inflated, artifact-heavy data. We apply a uniform preprocessing pipeline to all models: columns with excessive zeros or stuck sensors are dropped; short gaps are imputed; details in Appendix A.3.

**Recency Bias** Standard early stopping exhibits a recency bias: temporal proximity between train and validation sets causes expressive models to achieve deceptively strong validation scores in early epochs, triggering premature checkpoint selection. We observe that validation loss can be *anti-predictive* of test performance. For example, FERN's validation MSE worsens from 8.24 to 12.10 between epochs 1 and 6, yet test MSE improves from 18.88 to 5.51 over the same period (see Table A.4 for other models). To mitigate this, we apply a 3-epoch grace period and smoothed validation objective uniformly across all models.

## 4 EXPERIMENTS

Table 1 evaluate FERN against five LTSF baselines and one invariant learner: DLinear Zeng et al. (2023), TimeMixerWang et al. (2024), PatchTSTNie et al. (2023), Koopa Liu et al. (2023), ModernTCN Luo & Wang (2024) and PFNN Cheng et al. (2025) on 21 synthetic dynamical systems and 4 real-world benchmarks using 336-step input and 336-step forecast horizons, averaging results over 2 random seeds. Table 2 presents more comprehensive tests: a 4 random seed, 4 prediction horizons (96, 192, 336, 720) averaged results on FERN and TimeMixer, PatchTST and DLinear with standard errors on MSE. Setting details and system parameters are provided in AppendixA.7, and detailed table of comprehensive tests are in AppendixB.

### 4.1 EIGEN-PROFILE

For practitioners, prediction scoreboard is only half the story. **The arguably more important aspect for practitioners is knowing when to trust the prediction.** Here we illustrate how FERN provides *its own uncertainty belief*. Fig. 3 uses Lorenz-63 diagnostics to show how to utilize the *eigen-profile* when a flattened trajectory with low MSE (1/8 of the standard deviation) provides little insight. The eigen-structure reveals **where the model struggles**: around violent "top–bottom–top" flips of the trajectory, *both* the largest absolute error and the **maximum eigenvalue** (spectral radius) spike, meaning the model "believes" these segments to be the hardest, and the errors *confirm* this belief.

The **sum of eigenvalues** (trace) indicates total stretching. Trace is highest in relatively stable parts where the spectral radius is low, suggesting that less dominant eigen-directions are primarily activated when the Lorenz system traverses stably within a single lobe. This is consistent with HAVOK (Brunton et al., 2017), where less important directions of the *system* govern the intermittent forcing that 'randomly' propels the 'stable' trajectory from one lobe to another. Making this link precise is left to future work.

Finally, **log determinant** captures the *volume scaling* of the local map. The predominantly negative log-determinant (blue throughout) confirms FERN respects Lorenz's dissipative structure. At dark

| Dataset | fr MSE | fr WD | tm MSE | tm WD | tst MSE | tst WD | dl MSE | dl WD | kp MSE | kp WD | mtcn MSE | mtcn WD | pfnn MSE | pfnn WD |
|---|---|---|---|---|---|---|---|---|---|---|---|---|---|---|
| *Standard Chaotic Dynamics* | | | | | | | | | | | | | | |
| Rossler-Base | **0.019** | **0.011** | 1.03 | 0.903 | 2.45 | 2.25 | 5.42 | 5.06 | 11.94 | 5.58 | 0.47 | 0.42 | 21.05 | 16.64 |
| Rossler-Param | **0.036** | **0.017** | 3.49 | 2.91 | 10.02 | 8.62 | 28.74 | 25.42 | 25.07 | 17.91 | 1.64 | 1.37 | 28.09 | 23.08 |
| Lorenz-Base | **21.66** | **4.41** | 43.21 | 10.09 | 38.89 | 10.56 | 76.55 | 39.34 | 95.50 | 11.05 | 26.02 | 5.96 | 198 | 120 |
| Lorenz-State | **19.26** | **3.73** | 48.81 | 10.22 | 40.71 | 10.90 | 70.36 | 35.58 | 97.85 | 13.52 | 28.49 | 5.68 | 210 | 122 |
| Lorenz-Param | **25.21** | **4.61** | 52.10 | 10.63 | 40.59 | 9.19 | 70.69 | 32.89 | 103 | 17.86 | 35.96 | 7.57 | 219 | 151 |
| Lorenz96-Base | **5.19** | **1.33** | 8.03 | 2.97 | 6.35 | 2.49 | 10.98 | 6.02 | 17.42 | 3.41 | 10.38 | 3.64 | 20.17 | 15.56 |
| Lorenz96-Switch | **9.56** | **3.14** | 11.96 | 5.34 | 10.73 | 4.73 | 13.68 | 7.80 | 21.61 | 4.59 | 11.78 | 5.25 | 24.42 | 17.90 |
| Chua-Base | 0.056 | 0.033 | 0.094 | 0.046 | 0.186 | 0.119 | 0.720 | 0.507 | 1.11 | 0.482 | **0.051** | **0.029** | 1.77 | 1.67 |
| Chua-Param | 0.021 | 0.011 | **0.013** | **0.008** | 0.097 | 0.078 | 0.681 | 0.559 | 0.944 | 0.353 | 0.030 | 0.021 | 2.31 | 2.14 |
| Chua-Switch | 0.178 | 0.106 | 0.174 | 0.123 | 0.318 | 0.230 | 0.770 | 0.591 | 1.11 | 0.584 | **0.099** | **0.068** | 1.74 | 1.64 |
| *Switching Linear Dynamical System* | | | | | | | | | | | | | | |
| SLDS-Base | 2.84 | 1.46 | 4.54 | 2.80 | 2.27 | **1.05** | 4.42 | 3.50 | 2.96 | 1.96 | **1.96** | 1.10 | 3.52 | 2.97 |
| SLDS-Param | 2.36 | 1.36 | 2.60 | 1.58 | **2.18** | **0.91** | 2.26 | 1.50 | 2.19 | 1.38 | 2.30 | 1.80 | 2.57 | 2.13 |
| SLDS-Switch | **4.05** | **2.02** | 7.84 | 4.84 | 9.56 | 5.19 | 4.73 | 3.38 | 4.56 | 3.38 | 9.47 | 5.82 | 8.23 | 6.81 |
| *Seasonal AR Shocks (SAR)* | | | | | | | | | | | | | | |
| SAR-Base | **0.055** | 0.011 | **0.055** | 0.013 | 0.074 | 0.021 | 0.056 | **0.010** | 0.056 | 0.013 | **0.055** | 0.013 | 1.85 | 1.21 |
| SAR-Param | **0.355** | **0.053** | 0.361 | 0.065 | 0.480 | 0.128 | 0.380 | **0.053** | 0.366 | 0.075 | 0.359 | 0.065 | 10.83 | 7.43 |
| *GARCH Volatility Shocks* | | | | | | | | | | | | | | |
| GARCH-Base | **0.227** | 0.220 | 0.234 | 0.201 | 0.271 | **0.174** | 0.264 | 0.190 | 0.255 | 0.217 | 0.261 | 0.210 | 0.255 | 0.208 |
| GARCH-Param | **0.177** | 0.174 | 0.186 | 0.156 | 0.220 | 0.138 | 0.183 | **0.135** | 0.191 | 0.172 | 0.206 | 0.163 | 0.236 | 0.196 |
| *Double-Well Potential (DW)* | | | | | | | | | | | | | | |
| DW-Base | 0.054 | **0.028** | 0.059 | 0.043 | 0.091 | 0.057 | 0.055 | 0.034 | **0.049** | 0.042 | **0.049** | 0.043 | 1.47 | 1.41 |
| DW-Param | **0.682** | **0.506** | 1.08 | 0.843 | 0.983 | 0.721 | 0.847 | 0.711 | 1.03 | 0.856 | 1.00 | 0.815 | 0.837 | 0.756 |
| *Ornstein–Uhlenbeck (OU) Diffusions* | | | | | | | | | | | | | | |
| OU-Base | **0.234** | 0.195 | 0.251 | 0.131 | 0.273 | **0.112** | **0.234** | 0.166 | 0.251 | 0.156 | 0.241 | 0.178 | 0.237 | 0.216 |
| OU-Param | **0.239** | 0.170 | 0.251 | 0.131 | 0.273 | **0.112** | **0.239** | 0.163 | 0.251 | 0.156 | 0.241 | 0.178 | 0.383 | 0.358 |
| *Real-World Benchmarks* | | | | | | | | | | | | | | |
| ETTm1 | 8.97 | **5.37** | 9.12 | 5.63 | **8.83** | 5.49 | 9.71 | 6.25 | 9.22 | 5.69 | 11.14 | 7.11 | 43.69 | 34.25 |
| ETTm2 | 13.06 | 8.59 | 12.00 | **7.89** | 12.12 | 8.17 | 12.20 | 8.19 | **11.94** | 8.31 | 13.70 | 9.38 | 289 | 244 |
| ETTh1 | 10.97 | 5.75 | 10.51 | 5.23 | 10.97 | **4.83** | **10.39** | 5.00 | 10.75 | 5.55 | 14.52 | 9.01 | — | — |
| ETTh2 | 21.96 | 18.37 | 16.95 | 11.44 | 17.57 | **10.99** | 16.85 | 11.03 | **16.55** | 11.38 | 20.62 | 13.26 | — | — |

Table 1: **Stress-testing chaotic and stochastic systems with controlled non-stationarity.** 3 digits reported for < 1 entries, 2 digits for 1–100, integers for > 100. Models: fr (FERN), tm (TimeMixer), tst (PatchTST), dl (DLinear), kp (Koopa), mtcn (ModernTCN), pfnn (PFNN). Lower is better. **Purple**: best; Light Purple: second-best; Light Orange: diverged; '—': catastrophic.

blue regions, where the model commits to strong volume contraction, errors are lowest (pale, row 2). Errors tend to increase at lighter blue / white regions, suggesting that it finds expansive regimes more challenging.

## 4.2 WHAT HAVE WE LEARNED?

**Analysis** FERN achieves remarkable success on synthetic dataset experiments: in 19 out of 21 experiments FERN has best MSE or WD. The impressive 98% lower MSE than TimeMixer on Rössler, and the highly competitive results on stochastic and non-stationary benchmarks, suggests that the geometry-aware model is a well-calibrated, *all-around, 'safe'* model. The trend continues in the main tests. One *notable result* is in Table 14: At 720 steps (6.5 Lyapunov times given Lorenz-63's Lyapunov time is about 1.1 and $dt = 0.01$), pointwise prediction is provably impossible–errors amplify by $e^{6.5} \sim 650$. All models eventually converge to mean-guessing, but *when* matters: DLinear at horizon 96, TimeMixer/PatchTST at 192, FERN at 720. Notice More telling: even at 720, FERN's SWD is 4.89 versus 10–40 for others. This suggests (i) FERN's failure pattern against *the impossible task* is more robust while DLinear's is considerably worse than expressive models; (ii) FERN's geometry-awareness *is* important: between post-Lyapunov time and pre-mixing, geometry accuracy is still achievable when pointwise accuracy fails.

| Data | fr MSE | MAE | WD | tm MSE | MAE | WD | tst MSE | MAE | WD | dl MSE | MAE | WD |
|---|---|---|---|---|---|---|---|---|---|---|---|---|
| Lorenz | **21.82**± 2.13 | **2.17** | **2.23** | 30.94± 5.62 | 3.19 | 11.11 | 30.11± 2.92 | 3.28 | 9.60 | 67.76± 1.12 | 6.07 | 38.22 |
| Rossler | **0.04**± 0.01 | **0.11** | **0.02** | 6.01± 0.26 | 1.09 | 5.20 | 8.33± 0.36 | 1.43 | 7.25 | 11.64± 0.45 | 1.82 | 10.20 |
| Chua | **0.08**± 0.13 | **0.08** | **0.05** | 0.20± 0.21 | 0.16 | 0.15 | 0.49± 0.13 | 0.32 | 0.37 | 0.39± 0.02 | 0.30 | 0.24 |
| ETTm2 | **13.57**± 0.88 | **2.43** | **9.08** | 15.04± 0.76 | 2.50 | 10.45 | 15.63± 0.74 | 2.54 | 11.39 | 15.49± 0.63 | 2.55 | 10.98 |
| ETTh2 | 17.41± 0.75 | 2.82 | 11.12 | **14.20**± 0.82 | **2.46** | 8.54 | 14.44± 0.66 | 2.50 | 8.47 | 14.48± 0.60 | 2.49 | **7.92** |
| ETTh1 | **6.60**± 0.11 | 1.53 | **2.64** | 6.83± 0.16 | **1.52** | 2.83 | 6.62± 0.14 | 1.54 | 2.77 | 7.04± 0.06 | 1.53 | 2.75 |
| ETTm1 | 5.80± 0.25 | 1.45 | 2.85 | **5.27**± 0.22 | 1.39 | 2.60 | 5.36± 0.42 | **1.37** | **2.44** | 6.31± 0.11 | 1.39 | 3.65 |
| Wea* | 0.27± 0.02 | 0.30 | 0.22 | 0.27± 0.04 | 0.26 | 0.22 | 0.24± 0.01 | **0.24** | 0.20 | **0.21**± 0.00 | **0.24** | **0.16** |

Table 2: Aggregated errors across prediction horizons $H \in \{96, 192, 336, 720\}$ with input length=336. Baselines: **fr** = FERN, **tm** = TimeMixer, **tst** = PatchTST, **dl** = DLinear. Values are means over 4 seeds [7, 1955, 2023, 4]; tiny ± indicates standard error computed from per-horizon standard errors (see text). *Weather is normalized. Best and second-best values across models for each row/metric are highlighted via **a**nd .

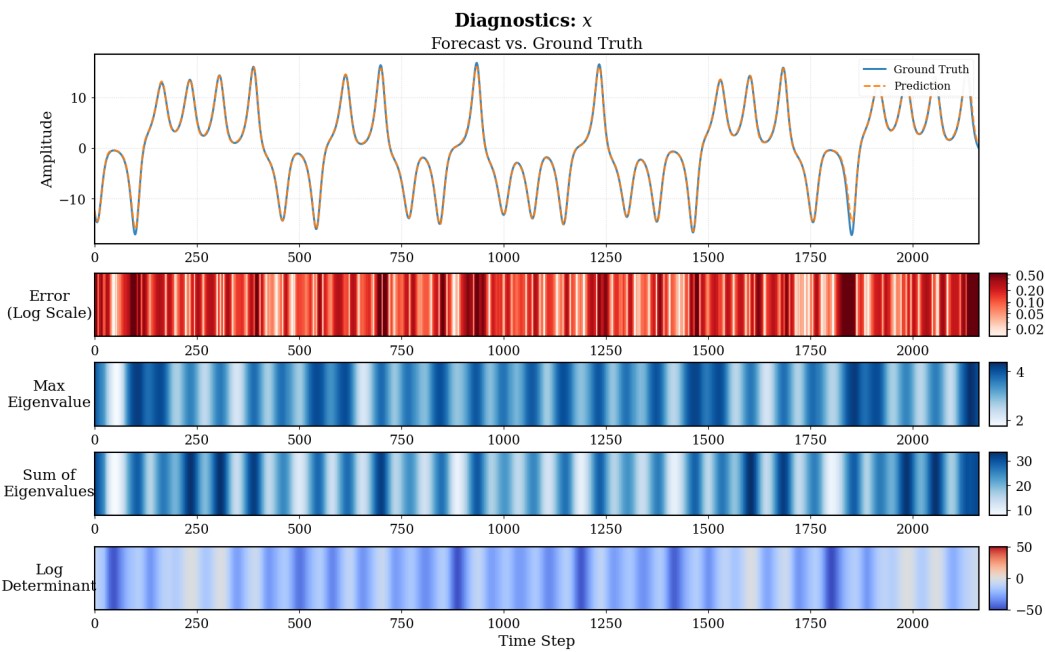

Figure 3: Diagnostics for Lorenz-63 x. Top to bottom: forecast (orange) vs. ground truth (blue); log absolute error; mean patch-wise max eigenvalue; mean patch-wise sum of eigenvalues; mean patch-wise log of products of eigenvalues.

More importantly, these results cast doubt on treating ETT as the *sole* benchmark and on the trend of elevating extremely simple models (DLinear and Koopa here) to *general-purpose* status based primarily on ETT. This risks rewarding brittle models. While feasible on *stochastic tasks and ETT*, on most performant datasets their advantages are about 20-30% over other models; Yet, the error explodes 5x-10x compared to TimeMixer under simple chaotic regime Rössler. As for PFNN, the numerical instability suggests that invariant learners currently struggle to compete on pointwise *conditional* prediction tasks. We discuss this in detail in Appendix A.2.2.

**The Spectrum Hypothesis** The ETT results in the shock test reveal an interesting pattern: the best models (PatchTST, DLinear, Koopa) *never* rank 1st place in chaotic tests; Yet the second strongest model in the synthetic tests, ModernTCN, however struggles. This suggests a *hypothesis* that the best ETT models, perhaps due to a strong smoothing component, excel in the 'low signal, high

| Variant (dataset) | MSE↓ | MAE↓ | WD↓ | EPT↑ | Time (s) |
|---|---|---|---|---|---|
| Base (ETTh2) | 21.96 | 3.27 | 18.37 | 129.87 | — |
| Base (ETTh1) | 10.96 | 1.88 | 5.75 | 63.34 | — |
| Base (Lorenz63) | **21.66** | **2.39** | **4.41** | 241.25 | — |
| No encoder & no mean updates (ETTh2) | 358.97 | 16.61 | 357.10 | 0.70 | 299.32 |
| No encoder & no mean updates (Lorenz63) | 194.43 | 10.99 | 189.47 | 17.10 | 492.73 |
| Only encoder (ETTh2) | 19.40 | 3.07 | 15.86 | 137.20 | 176.97 |
| Only encoder (ETTh1) | 11.17 | 1.87 | 5.86 | 63.00 | 378.93 |
| Only encoder (Lorenz63) | 27.09 | 2.85 | 6.17 | 214.10 | 1127.98 |
| No rotation (ETTh2) | 27.80 | 3.46 | 17.20 | 120.00 | 224.05 |
| No rotation (ETTh1) | 11.84 | 1.90 | 7.72 | 66.00 | 617.55 |
| No rotation (Lorenz63) | 27.62 | 2.94 | 6.19 | 203.60 | 846.38 |
| No patching (ETTh2) | 20.14 | 3.15 | 17.26 | 137.60 | 184.73 |
| No patching (ETTh1) | 10.99 | **1.84** | 5.38 | 64.10 | 422.65 |
| No patching (Lorenz63) | 22.86 | 2.50 | 4.47 | 231.10 | 1239.11 |
| Reflection = 2 (ETTh2) | 19.34 | **2.99** | **14.90** | 139.40 | 186.72 |
| Reflection = 2 (ETTh1) | 11.52 | 1.89 | 6.02 | 64.30 | 429.67 |
| Reflection = 2 (Lorenz63) | 26.14 | 2.79 | 6.51 | 199.10 | 889.23 |
| Reflection = 24 (8-block) (ETTh2) | **18.57** | 3.06 | 14.99 | **141.50** | 197.86 |
| Reflection = 24 (8-block) (ETTh1) | **10.91** | 1.87 | **5.21** | 63.00 | 432.86 |
| Reflection = 24 (8-block) (Lorenz63) | 23.92 | 2.70 | 5.70 | 213.00 | 930.42 |

Table 3: Ablations of FERN components at prediction length 192 on ETTh2, ETTh1, and Lorenz-63. Base uses reflection=8 and patch size P=24; variants change exactly one component at a time. We report MSE, MAE, Wasserstein distance (WD), effective prediction time (EPT; higher is better), and wall-clock training time per run. **Bold** marks the best (lowest) MSE, MAE, or WD for each dataset across all variants. Baseline runtimes were not recorded and are shown as '—'.

noise' spectrum where data artifacts and real-world noises are substantial. At the other end of the spectrum, characterized by *clean signals and reasonable system noise*, lie FERN and ModernTCN, which score best on true seasonal AR processes but only competitively on ETT. A wise combination of models in practice seems prudent.

### 4.3 ABLATION

We provide extensive ablations in Table 3. We find: (i) without encoder, the model is useless (MSE and SWD explode on both ETTh2 and Lorenz), suggesting that the bidirectional encoder successfully compresses the input information into the latent space; (ii) in the 'only encoder' case, we disable the repeated translation in the decoder. This slightly helps ETTh2 but hurts ETTh1 and, more strongly, Lorenz, suggesting that the data-dependent translations are particularly important for chaotic systems, while on noisy ETTh2 (plagued by data-quality issues) this extra flexibility only partly translates into MSE gains; (iii) removing either the eigen-based rotation or the patching mechanism consistently degrades Lorenz performance and often harms ETTh2/ETTh1, confirming that local geometric alignment and patch-wise conditioning are both doing real work; and (iv) increasing the number of reflections from 0 (no rotation) to 24 **monotonically** improves MSE, SWD, and EPT on all three datasets at a modest runtime cost, making the 24-reflection configuration the strongest variant in this ablation grid. This suggests that a full-complexity spectral map *is* important, and indirectly confirms the importance of patching, which makes a low-cost search possible.

## 5 DISCUSSION AND RELATED WORK

The LTSF field has been dominated by Transformer variants (Zhang & Yan, 2023; Liu et al., 2024). Foundational critiques (Zeng et al., 2023; Bergmeir, 2023) on model complexity and evaluation paradigms catalyzed a shift to simplicity and interpretability. Recent efforts emphasize linearity and efficiency (Xu et al., 2024; Zhou et al., 2025; Huang et al., 2025), and frequency-domain analysis for periodic signals (Wang et al., 2025; Piao et al., 2024; Wu et al., 2023). Another active area is

**Koopman operator theory** (Brunton et al., 2022), which promises to *linearize* nonlinear dynamics by lifting states into a space where a linear operator governs evolution (Liu et al., 2023; Zhang et al., 2025). Our 1D W2 proposal coincides with similar ideas that have been developed independently in several fields (Muskulus & Verduyn Lunel, 2011; Wiesel, 2022; Aoun et al., 2024; Botvinick-Greenhouse et al., 2024), suggesting a convergent consensus.

## 5.1 Why Fern Works

Fern is different from traditional models:

1. it is a *transformation-generating network*: while the underlying network is nonlinear, black-box and opaque, the *applied transformation* is stable, smooth, well-behaved with exact geometric meaning. This eliminates a large class of singular peculiarities with general MLP. Within this class, since Householder transformation is orthogonal and numerically stable, the true source of problems is the eigenvalues for which users can easily tune. For example, we apply a 4.5 upper bound for the eigenvalues;

2. for DGP that is deterministic plus heteroskedasticity, i.e. varying non-constant Gaussian errors, the ellipsoid method *may* be pushing the pointwise error toward its Bayes floor subject to network capacity, thanks to its fully data-driven nature.

3. we view real-world nonstationary shocks as **piecewise-ergodic regimes**: within a regime, the conditional/invariant geometry is stable; shocks trigger switches. Each regime leaves a **geometrically distinct distributional signature**. The forecaster's task, is to detect and track these shifts: widening ellipsoids to express uncertainty during anticipated changes, then confining predictions to regime-specific compact ellipsoids post-switch. Pointwise error becomes a byproduct of this process, as learning regime-specific geometries is more tractable than recovering governing dynamics under chaos and noise.

## REPRODUCIBILITY STATEMENT

We provide an anonymous code link https://anonymous.4open.science/status/FERN-1F63 for illustration and submit them in the supplemental materials. Full implementation will be released upon paper acceptance. Appendix B details individual runs over 4 seeds with standard error for each model, and average performances; Algorithm 1 outlines the entire FERN structure; Appendix A.6 lists datasets and preprocessing; Appendix 15 gives hyperparameters, seeds, and general details of the experiment; Appendix A.5 list the notations, theorems and definitions and technical details.

## ETHICS STATEMENT

We have no ethical concerns to report.

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

## A APPENDIX

### A.1 LLMs USAGE

We used GPT- and Gemini-class LLMs for table/figure layout, literature-search support, editing, and LaTeX formatting; All LLM outputs were reviewed and verified by the authors, who remain fully responsible for the content. All scientific content, including the model structure, math connection, intuition, are entirely authors' own and sharpened during (sometimes intense) brainstorming with AI. Specifically, we thank GPT and GEMINI for their contribution to this project, and their developers for making them available.

### A.2 EXTRA DISCUSSION

#### A.2.1 FERN'S ROLE, CLARIFIED

**What FERN is, and what FERN is not**  Due to frequent misunderstanding about FERN's role as a forecast model, we clarify: FERN is **not an iterated one-step chaos specialist.** Such models recursively feed their own output to simulate long trajectories, and chaos-tuned variants (e.g., DSDL) excel on Lorenz dataset (Wang & Li, 2024). FERN is instead a direct, multi-step forecaster for general-purpose tasks, stress-tested on chaotic datasets only as a supplement to standard LTSF benchmarks. Also, the metric EPT we advocate for, is a common measure of one-step *model stability* in chaotic dynamics, alongside variants like VPT (valid prediction time). We use EPT to measure *dataset predictability*. Direct comparison between iterated and multi-step models can be misleading, as the latter are not optimized for it.

FERN is **not a governing-equation or invariant-measure learner**. It aims at discrete-time conditional forecasting with data-driven spectral factors rather than system identification, distinct from (i) sparse governing equation finding SINDy/HAVOK/VINDy (Brunton et al., 2016; 2017; Conti et al., 2024) (ii) Neural-ODE identification (Ko et al., 2023) (iii) invariant-measure learner (Jiang et al., 2023; Schiff et al., 2024).

#### A.2.2 MARKOV NEURAL OPERATORS

Operator-learning approaches (PFNN, MNO **?**) are *invariant measure* learners of chaotic regimes, which pursue a complementary agenda: learning global evolution operators for specific systems. In contrast, FERN targets local conditional geometry for generic forecasting tasks, prioritizing per-window spectral diagnostics over explicit operator recovery.

However, many requests insist on probing FERN's pointwise performances vs this class, and so we complied. In the main text, Table. 1 suggests PFNN performs reasonably on some smooth diffusion tasks (OU, DW), but often exhibits very large errors or outright divergence on chaotic and heavily forced settings (e.g., Rössler, Lorenz, ETTm1/2), under the unified training protocol. In the worst ETTh2 case, the error explodes into billions.

We were initially concerned about the errors and perform additional hyperparameter (learning rate, latent dimension size, network architecture etc) searches. Our implementation carefully follows the PFNN public repository. Based on (Cheng et al., 2025), the reported NRMSE is $0.49$ on $\sim 80000$ samples with $dt = 10^{-2}$ for Lorenz–63. Since the standard deviation of the Lorenz system is about 8, this corresponds to roughly 15–20 MSE over a 100-step rollout. In contrast, FERN attains an MSE of about 1.2 on 22k samples with a 96-step prediction horizon. We conclude that the reported error on 336-step rollout is likely consistent with the paper.

**Why the difference?  Markov System vs Conditional Forecasting.**  Conceptually, LTSF models take a history window of shape $[\text{batch}, \text{feature}, \text{seq}_{0:100}]$ and predict a future window $[\text{batch}, \text{feature}, \text{seq}_{100:200}]$, often with *channel-independent* parametrization, whereas Markov(-type) operators act along the temporal axis, mapping $[\text{batch}, \text{seq}_{0:1}, \text{feature}]$ to $[\text{batch}, \text{seq}_{1:2}, \text{feature}]$ by evolving the state one step at a time using *only* the *current state* information.

Lorenz etc. are **NOT** Markov systems; the past information informs the model about its likely current location. The task of *MNO* is to *learn the global invariant measure* based on *current* state,

which is a **strictly more difficult problem** than *learning conditional forecast distribution* with recent information. Lorenz63 has 3 state variables, $\sim 10$ variables are enough to invoke Takens' Embedding Theorem, so the remaining input length serves as historical context. We view this not as a flaw in PFNN per se, but as evidence that architectures tuned for Markovian operator learning do not transfer automatically to windowed multi-horizon forecasting; conversely, FERN does not currently target PDE operators and should not be read as a replacement in that regime.

## A.3 DATA PROTOCOL

**Issues with ETTh2**   In contrast to synthetic data's cleanness, real-world data has data quality issues. We illustrate how the zeros affecting ETTh2 in Fig. 4. The frequent appearance of clusters of zeroes plague not just on spot, but *every* rolling window containing that data. It treat the model's correct, stable predictions as *entirely wrong*. Models that responds with *lower averages* are significantly rewarded; on the other hand, robust models are penalized, making the MSE metrics uninformative.

Table. 4 analyze the pattern of zeroes; we discover that:

1. As Table. 4 suggests, several columns are heavily zero-inflated: the two columns 'HULL' and 'LULL' has 22% and 33% zeroes respectively. Standard pointwise losses such as MSE, MAE (i.e. Gaussian / Laplace error assumption) are *not* equipped to handle mixture of continuous and structural' zeros.

2. *Long consecutive zeros* plague the data. Worse, sentinel values like 88.297 stuck and persist for months, which is almost certainly data processing artifacts than physical reality.

**Pre-processing Scheme**   Our fix is policy based: (i) if known sentinel values ¿10% or artificial zeros ¿ 15%, we drop the column for poor data quality. If 10% ¡ zeros ¡ 15%, we apply asinh transform to stabilize heavy tails while preserving slope around zero; (ii) If the *longest* zero run in a column exceeds a week we drop that column: the sensor is effectively stuck. Below that and 3 hours, we delete the rows affected by long zeros; below that, we impute with forward–backward fill.

Table 4: Zero–inflation patterns in ETTH2. "Clustered" counts zeros that belong to runs of length $> 1$ (consecutive zeros). Type 1 columns show substantial intermittent zero bursts; Type 2 suggests occasional missing entries.

| Column | Total Zeros | % Zeros | Isolated | Clustered |
|--------|------------|---------|----------|-----------|
| HUFL   | 58         | 0.33%   | 1        | 57        |
| HULL   | 3836       | 22.02%  | 163      | 3673      |
| MUFL   | 0          | 0.00%   | 0        | 0         |
| MULL   | 1067       | 6.13%   | 245      | 822       |
| LUFL   | 1188       | 6.82%   | 184      | 1004      |
| LULL   | 5792       | 33.25%  | 414      | 5378      |
| OT     | 1          | 0.01%   | 1        | 0         |

Figure 4: Reconstruction of ETTh2's HULL column: multiple consecutive zeros severely plague the dataset

## A.4 CHECKPOINT STRATEGY

**Why a grace period?**   Early validation error in long-horizon forecasting can be *anti-predictive* of final test error. Empirically (also mentioned in Zeng et al. (2023)), with a naive early stopping protocol, expressive models (e.g., PatchTST, TimeMixer, FERN) achieve a deceptively strong validation score in the initial epochs. This often "lock in" expressive models to their first epoch weights, trapping a model in a poor local minimum. Another remedy essential to FERN (and other expressive models, but to a less extent) is *exponential smoothing* of validation objective: we observe the fully data-driven model often oscillate between overfitting and generalized solutions during training, a 3-epoch smoothing is sufficient to incorporate progresses in *both* generalized and overfitting solutions.

In this study, an earlier version of FERN was used. For most expressive models, the test MSE at a later epoch (e.g., Epoch 6) is significantly better than at Epoch 1, even when the corresponding validation MSE is worse. Therefore, early validation loss is an unreliable proxy for generalization performance, except for less expressive models like DLinear, which are less flexible.

Table 5: Validation drift and test improvement (MSE). $\Delta$Val = Val@later − Val@1; $\Delta$Test = Test@later − Test@1. Positive $\Delta$Val means validation worsens after epoch 1; negative $\Delta$Test means later epochs improve test error.

| Model–Data | Val@1 | Best Val | Ep | $\Delta$Val | Test@1 | Best Test | Ep | $\Delta$Test |
|---|---|---|---|---|---|---|---|---|
| TST–ETTh1 | 6.49 | 9.24 | 6 | +2.75 | 6.85 | 6.21 | 6 | -0.64 |
| TST–ETTm2 | 4.45 | 5.99 | 10 | +1.54 | 14.58 | 12.08 | 10 | -2.50 |
| TimeMixer–ETTh1 | 6.28 | 9.51 | 6 | +3.23 | 7.38 | 7.34 | 6 | -0.04 |
| TimeMixer–ETTm2 | 6.03 | 10.17 | 7 | +4.14 | 13.37 | 13.22 | 7 | -0.15 |
| FERN–ETTh1 | 8.24 | 12.10 | 6 | +3.86 | 18.88 | 5.51 | 6 | -13.37 |
| FERN–ETTm2 | 58.43 | 16.09 | 6 | -42.34 | 78.30 | 18.71 | 6 | -59.59 |
| DLinear–ETTh1 | 7.12 | 6.22 | 6 | -0.90 | 7.79 | 7.39 | 6 | -0.40 |
| DLinear–ETTm2 | 5.21 | 5.49 | 8 | +0.28 | 15.66 | 12.87 | 8 | -2.79 |

## A.5 BACKGROUND DEFINITIONS, NOTATIONS, AND THEOREMS

**Householder-based rotation parameterization.** Any orthogonal matrix $U \in \mathbb{R}^{p \times p}$ can be written as a product of at most $p$ Householder reflections (Householder, 1958; Golub & Van Loan, 1996):

$$H_i := I - 2v_i v_i^\top, \qquad \|v_i\|_2 = 1,$$

where each $H_i$ is symmetric with $\det(H_i) = -1$ and reflects vectors across the hyperplane orthogonal to $v_i$. Composing $R$ reflections,

$$U = H_R H_{R-1} \cdots H_1,$$

yields an orthogonal matrix; when $R$ is even, $\det(U) = +1$ (a proper rotation) and inversion is just transposition $U^{-1} = U^\top$. Applying one reflection to a vector costs $O(p)$ (compute $v_i^\top x$, scale $v_i$, subtract), so $R$ reflections cost $O(Rp)$ and require $O(Rp)$ parameters if each $v_i$ is produced by an MLP. Taking $R = p$ recovers the full orthogonal group; using $R \ll p$ gives a low-rank, data-dependent approximation to the rotation part of the transport Jacobian at linear cost in horizon.

**Takens' embedding (Takens, 1981).** Let $M$ be a compact $d_M$-dimensional manifold, $\phi : M \to M$ a diffeomorphism, and $y : M \to \mathbb{R}$ a smooth observable. For $m \geq 2d_M + 1$, the delay map

$$\Phi(x) = \left(y(x), y(\phi^{-1}(x)), \ldots, y(\phi^{-(m-1)}(x))\right) : M \to \mathbb{R}^m$$

is generically an embedding. Thus the reconstructed attractor in $\mathbb{R}^m$ is diffeomorphic to the original one on $M$, preserving topology and dynamics.

**Brenier–McCann (quadratic optimal transport) (Brenier, 1991).** Let $\mu, \nu$ be Borel probability measures on $\mathbb{R}^n$ with finite second moments, and assume $\mu$ is absolutely continuous w.r.t. Lebesgue measure. For cost $c(x, y) = \frac{1}{2}\|x - y\|^2$, there exists a unique optimal transport map $T : \mathbb{R}^n \to \mathbb{R}^n$ pushing $\mu$ to $\nu$ $(T_\# \mu = \nu)$ of the form $T = \nabla \psi$ for a convex potential $\psi$.

**Wasserstein–2 distance** Let $\mathcal{P}_2(\mathbb{R}^n)$ be the set of Borel probability measures on $\mathbb{R}^n$ with finite second moment. For $\mu, \nu \in \mathcal{P}_2(\mathbb{R}^n)$ and quadratic cost $c(x, y) = \|x - y\|^2$,

$$W_2^2(\mu, \nu) = \inf_{\gamma \in \Pi(\mu, \nu)} \int_{\mathbb{R}^n \times \mathbb{R}^n} \|x - y\|^2 \, d\gamma(x, y),$$

where $\Pi(\mu, \nu)$ is the set of all couplings (transport plans) with marginals $\mu$ and $\nu$.

**Monge map (when it exists)**   If the infimum is attained by a measurable map $T : \mathbb{R}^n \to \mathbb{R}^n$ with $T_{\#}\mu = \nu$ (a Monge solution), then

$$W_2^2(\mu, \nu) \;=\; \int_{\mathbb{R}^n} \|x - T(x)\|^2 \, \mathrm{d}\mu(x).$$

Under Brenier's conditions (e.g., $\mu$ absolutely continuous w.r.t. Lebesgue), such a map exists and is unique a.e., with $T = \nabla\psi$ for a convex potential $\psi$.

**Pushforward**   For measurable $T : \mathbb{R}^n \to \mathbb{R}^n$ and a measure $\mu$, the pushforward $T_{\#}\mu$ is

$$T_{\#}\mu(B) \;=\; \mu\big(T^{-1}(B)\big) \qquad \text{for all Borel sets } B \subseteq \mathbb{R}^n.$$

Equivalently, $\int f(y) \, \mathrm{d}(T_{\#}\mu)(y) = \int f(T(x)) \, \mathrm{d}\mu(x)$ for all bounded measurable $f$.

**Sliced Wasserstein Distance (SWD)**   Given predictions and targets $\{y_b^{\text{pred}}\}_{b=1}^B, \{y_b^{\text{true}}\}_{b=1}^B \subset \mathbb{R}^H$, we view them as empirical measures $\mu = \frac{1}{B}\sum_{b=1}^B \delta_{y_b^{\text{pred}}}$ and $\nu = \frac{1}{B}\sum_{b=1}^B \delta_{y_b^{\text{true}}}$. The sliced Wasserstein–1 distance is

$$\mathrm{SW}_1(\mu, \nu) = \int_{S^{H-1}} W_1\big((\theta^{\top}\cdot)_{\#}\mu, \ (\theta^{\top}\cdot)_{\#}\nu\big) \, d\sigma(\theta),$$

where $S^{H-1}$ is the unit sphere, $\sigma$ the uniform surface measure, and $(\theta^{\top}\cdot)_{\#}\mu$ the pushforward of $\mu$ by $x \mapsto \theta^{\top}x$. For 1D empirical samples $U = \{u_b\}, V = \{v_b\}$ with equal weights,

$$W_1(U, V) = \frac{1}{B} \sum_{b=1}^B |u_{(b)} - v_{(b)}|,$$

with $u_{(b)}, v_{(b)}$ the order statistics. We approximate the sphere integral with $L$ random directions $\{\theta_\ell\}_{\ell=1}^L$:

$$\mathrm{SW}_1(\mu, \nu) \approx \frac{1}{L} \sum_{\ell=1}^L W_1\Big(\{\theta_\ell^{\top} y_b^{\text{pred}}\}_{b=1}^B, \ \{\theta_\ell^{\top} y_b^{\text{true}}\}_{b=1}^B\Big).$$

See Rabin et al. (2011); Bonneel et al. (2015); Kolouri et al. (2019). Importantly, when we write WD in the table, it means SWD without projection (which reduces to 1D WD) and when we write SWD in the table, it means SWD with 500 random projections.

### A.5.1   Effective Prediction Time (EPT)

EPT measures how long a forecast stays within a $1\sigma$ error envelope for chaotic systems. Let $y_{b,d,s}^{\text{pred}}$ and $y_{b,d,s}^{\text{true}}$ be predicted and true values for batch $b$, dimension $d$, step $s$ (horizon $H$). Let $\epsilon_d = \sigma_d$ be the standard deviation of the training data for dimension $d$. For each $(b, d)$ define

$$\mathrm{EPT}_{b,d} = \min\big\{s \in \{1, \ldots, H\} \,\big|\, |y_{b,d,s}^{\text{pred}} - y_{b,d,s}^{\text{true}}| > \epsilon_d\big\},$$

with $\mathrm{EPT}_{b,d} = H$ if the threshold is never exceeded. The reported score is the average

$$\mathrm{EPT}_{\text{avg}} = \frac{1}{B\,D} \sum_{b=1}^B \sum_{d=1}^D \mathrm{EPT}_{b,d}.$$

Thus EPT is the average number of forecast steps before the error exceeds one standard deviation; higher EPT is better (all other metrics are "lower is better").

### A.6   Systems and Datasets

### A.6.1   Chaotic Systems

Prior work shows that deterministic finite-precision arithmetic can suppress chaos and produce spurious periodic orbits in low precision, and that chaotic systems are highly sensitive to numerical

precision and discretization choices. See, e.g., Klöwer et al. (2023) on periodic orbits at low precision and mitigation via stochastic rounding; Teixeira, Reynolds & Judd (2007) on decoupling times and truncation-error growth; and Liao (2014) on the joint impact of truncation and round-off errors on long-time chaotic simulations. (Klöwer et al., 2023; Teixeira et al., 2007; Liao & Wang, 2014). Therefore, all chaotic ODE datasets (Lorenz63, Rössler, Chua, Lorenz96) were generated in `float64` using the 4th-order Runge-Kutta (RK4) method Butcher (1987) and then converted to standard `float32` Pytorch Tensor. Pilot runs observed materially larger forecast errors across all models with `float64`.

**Lorenz63.**   A canonical 3-D system modeling atmospheric convection:
$$\dot{x} = \sigma(y - x), \quad \dot{y} = x(\rho - z) - y, \quad \dot{z} = xy - \beta z,$$

**Rössler.**   A 3-D system with a folded-band attractor:
$$\dot{x} = -y - z, \quad \dot{y} = x + ay, \quad \dot{z} = b + z(x - c),$$

**Chua's Circuit**   A 3-D piecewise-linear circuit model with a double-scroll attractor:
$$\dot{x} = \alpha\big(y - x - h(x)\big), \quad \dot{y} = x - y + z, \quad \dot{z} = -\beta y,$$
with
$$h(x) = m_1 x + \tfrac{1}{2}(m_0 - m_1)\big(|x + 1| - |x - 1|\big).$$

**Lorenz96.**   A $d$-dimensional toy model for mid-latitude atmospheric dynamics, with cyclic nearest-neighbour coupling and constant forcing forcing $= F$ (parameter `forcing` in code, dimension `dim`):
$$\dot{x}_j = \big(x_{j+1} - x_{j-2}\big) x_{j-1} - x_j + F, \qquad j = 1, \ldots, d,$$
with indices taken modulo $d$ (e.g. $x_0 = x_d$, $x_{-1} = x_{d-1}$).

### A.6.2   STOCHASTIC SYSTEM

**Ornstein–Uhlenbeck (OU).**   A 1-D mean-reverting diffusion with linear drift towards a long-run mean $\mu$ and Gaussian noise, with parameters $(\theta, \mu, \sigma)$ (`theta`, `mu`, `sigma` in code):
$$dX_t = \theta\big(\mu - X_t\big)\, dt + \sigma\, dW_t,$$
integrated with an Euler–Maruyama scheme using step size `dt`.

**Double-well SDE.**   A 1-D bistable diffusion in a double-well potential, parameterized by a shape parameter $a$ and noise scale $\sigma$ (`a`, `sigma` in code):
$$dX_t = \big(aX_t - X_t^3\big)\, dt + \sigma\, dW_t.$$
The deterministic drift $aX_t - X_t^3$ creates two metastable wells around $\pm\sqrt{a}$, with noise-driven transitions between wells.

**Switching linear (SLDS).**   A 1-D switching linear dynamical system (SLDS) with two linear-Gaussian regimes, specified by $(A_1, Q_1)$ and $(A_2, Q_2)$ and Markov self-transition probabilities $p_{11}, p_{22}$ (`A1`, `Q1`, `A2`, `Q2`, `p11`, `p22` in code). Let $s_t \in \{1, 2\}$ be the latent regime:
$$x_{t+1} = A_{s_t} x_t + \eta_t, \qquad \eta_t \sim \mathcal{N}(0, Q_{s_t}),$$
$$\mathbb{P}[s_{t+1} = i \mid s_t = i] = p_{ii}, \quad i \in \{1, 2\}.$$
This yields piecewise-linear dynamics with regime switches driven by a 2-state Markov chain.

**Seasonal AR.**   A 1-D discrete-time process with an autoregressive term and an explicit seasonal component of period $S$ (`S`) and slowly drifting amplitude. With AR coefficient $\phi$ (`phi`), innovation scale $\sigma$ (`sigma`), initial seasonal amplitude $a_0$ (`a0`), and linear drift rate `amp_drift_per_step`, we write
$$a_t = a_0 + t \cdot \text{amp\_drift\_per\_step},$$
$$x_t = a_t \cos\!\Big(\tfrac{2\pi t}{S}\Big) + \phi\, x_{t-1} + \sigma\, \varepsilon_t, \qquad \varepsilon_t \sim \mathcal{N}(0, 1).$$
This models a gradually drifting seasonal pattern on top of an AR(1) background.

**GARCH(1,1).** A discrete-time volatility process with conditionally Gaussian returns and autoregressive conditional variance, parameterized by $(\omega, \alpha, \beta)$ (omega, alpha, beta in code):

$$x_t = \sigma_t \, \varepsilon_t, \qquad \varepsilon_t \sim \mathcal{N}(0, 1),$$
$$\sigma_t^2 = \omega + \alpha x_{t-1}^2 + \beta \sigma_{t-1}^2.$$

Here $x_t$ represents heavy-tailed, volatility-clustered "returns", while $\sigma_t^2$ evolves as a GARCH(1,1) variance process.

**Kuramoto–Sivashinsky (KS).(Unused in the paper)** A 1-D chaotic PDE with strong spatiotemporal instability, simulated on a periodic domain of length $L_x$ with $n_x$ grid points (Lx, nx) and viscosity $\nu$ (nu in code). In continuous form the field $u(t, x)$ satisfies

$$u_t + u\, u_x + u_{xx} + \nu\, u_{xxxx} = 0, \qquad x \in [0, L_x],$$

with periodic boundary conditions. We discretize in space using $n_x$ modes and evolve in time with an ETDRK4 scheme (method='etdrk4') and step size dt.

### A.6.3   REAL-WORLD BENCHMARKS

**ETT (Electricity Transformer Temperature).** Data from two Chinese electricity substations transformers over 2 years:

- ETTm1, ETTm2: 7 variables, 15 min sampling, 69,680 time steps.
- ETTh1, ETTh2: 7 variables, 1 h sampling, 17,420 time steps

**Weather**   21 meteorological indicators at 10 min intervals (Max Planck Institute, Germany, 2020).

### A.7   EXPERIMENTS SETUP

**Training setup**   The main experiments were run earlier than the shock experiments. Unless specifically noted, the experiment details are the same for both set of experiments. All models are implemented in PyTorch and trained with AdamW (Loshchilov & Hutter, 2019) (lr $3 \times 10^{-4}$, weight decay 0), batch size 95, for up to 50 epochs with patience $= 5$. We use a 3-epoch and 5-epoch *grace period* in shock and main test respectively: validation is logged but cannot trigger early-stopping during these periods. The training and is Huber loss ($\delta = 1.0$). We evaluate models using MSE, MAE, and SWD (to be clear, we do not use any projections, reducing SWD to 1D W2 loss explained in the paper), in particular, validating objective (for early stop) is 0.1·MSE + 1.0·MAE + 0.1·SWD; SWD is reported, not optimized.

The shock experiments and ablations are run on a {336} input length, {336} forecast horizon case on two seeds [7, 1955] with a **70%/20%/10% split** (to make reconstruction test set easier). In these experiments, automatic pre-processing is enabled for the real-world datasets. Specific sentinel codes are added to both ETTm2 and ETTh2. Validation is universally based on **best 3-epoch EMA** validation objective. 1D WD (SWD without projection) is used.

For main experiments, the prediction horizons are {96, 192, 336, 720}. Input length is {336}. We follow standard **70%/10%/20%** train/validation/test. All datasets are processed without any drop_last setting for loaders. In these experiments, automatic pre-processing is **not** enabled for the real-world datasets. No sentinel code fixes too, naturally. Validation is universally based on **best** validation objective. SWD with 500 projections is used.

### A.7.1   INTRO TO SHOCK EXPERIMENTS

We design three non-stationarities that **happen at *exactly* the midpoint of the training data:**

1. **Parameter drift:** system parameters change midway: e.g. Lorenz-63 (the famous butterfly effect system) Lorenz (1963) parameters were generated with $\sigma = 10, \rho = 28, \beta = 8/3$; halfway, parameters become $\sigma = 10.1, \rho = 28.1, \beta = 8.1/3$ and keep generating the rest. The system evolves into a Lorenz trajectory with *different* parameters but *same* initial conditions.

2. **State perturbation:** halfway, a specified shock such as $\epsilon = 0.9$ is added to the three state variables of Lorenz-63, and we keep generating the rest. Due to sensitive dependence on initial conditions, the system then evolves into a Lorenz trajectory with *the same* parameters but *different* initial conditions.

3. **Regime replacement:** halfway, the trajectory is replaced by another trajectory generated with *different* parameters and *different* initial conditions.

### A.7.2 DETAILED SYSTEM PARAMETER USED

| System | Scenario | $dt$ | steps | Parameters / shock description |
|---|---|---|---|---|
| *Main chaotic benchmarks (no shock).* | | | | |
| Lorenz-63 | main | 0.01 | 25000 | `sigma` = 10, `rho` = 28, `beta` = 8/3. |
| Rössler | main | 0.01 | 25000 | `a` = 0.2, `b` = 0.2, `c` = 5.7. |
| Chua | main | 0.005 | 35000 | `alpha` = 15.6, `beta` = 28.0, `m0` = −8/7, `m1` = −5/7. |
| *Synthetic shock scenarios (code identifiers* `PremadeID.*`*).* | | | | |
| Lorenz-63 | `LORENZ_BASE` | 0.01 | 35999 | Baseline Lorenz-63, no shock; default `sigma` = 10, `rho` = 28, `beta` = 8/3. |
| Lorenz-63 | `LORENZ_PARAM` | 0.01 | 35999 | Parameter shock (`shock_kind` = `"param"`): `sigma` : $10 \rightarrow 10.1$, `rho` : $28 \rightarrow 28.1$, `beta` : $8/3 \rightarrow 8.1/3$. |
| Lorenz-63 | `LORENZ_STATE` | 0.01 | 35999 | State shock (`shock_kind` = `"state_eps"`): `shock_eps` = 0.9; ODE parameters as in `LORENZ_BASE`. |
| Lorenz-63 | `LORENZ_SWITCH` | 0.01 | 35999 | Switch shock (`shock_kind` = `"switch"`): `switch_update` sets `rho` : $28 \rightarrow 28.1$ and `initial_cond` : $[1.0, 0.98, 1.1] \rightarrow [1.002, 0.982, 1.102]$. |
| Rössler | `ROSSLER_BASE` | 0.01 | 35999 | Baseline Rössler, no shock; `a` = 0.2, `b` = 0.2, `c` = 5.7. |
| Rössler | `ROSSLER_PARAM` | 0.01 | 35999 | Parameter shock (`shock_kind` = `"param"`): `a` : $0.2 \rightarrow 0.25$, `b` : $0.2 \rightarrow 0.25$, `c` : $5.7 \rightarrow 5.75$. |
| Lorenz-96 | `LORENZ96_BASE` | 0.007 | 55000 | Baseline Lorenz-96; `dim` = 6, `forcing` = 8.0, `method` = "rk4". |
| Lorenz-96 | `LORENZ96_SWITCH` | 0.007 | 55000 | Switch shock (`shock_kind` = `"switch"`): `switch_update` sets `forcing` : $8.0 \rightarrow 9.0$ and `initial_cond` = $[0.99, 1.02, 1.02, 1.03, 1.01, 1.01]$ (with `dim` = 6, `method`="rk4"). |
| Chua | `CHUA_BASE` | 0.005 | 35999 | Baseline Chua, no shock; `alpha` = 15.6, `beta` = 28.0, `m0` = −8/7, `m1` = −5/7. |
| Chua | `CHUA_PARAM` | 0.005 | 35999 | Parameter shock (`shock_kind` = `"param"`): `alpha` : $15.6 \rightarrow 15.9$, `beta` : $28.0 \rightarrow 28.5$, `m0` : $-8/7 \rightarrow -8.1/7$, `m1` : $-5/7 \rightarrow -5.2/7$. |
| Chua | `CHUA_SWITCH` | 0.005 | 35999 | Switch shock (`shock_kind` = `"switch"`): `switch_update` sets `initial_cond` = $[0.11, 0.01, 0.02]$; other parameters as in `CHUA_BASE`. |
| OU | `OU_BASE` | 0.5 | 25000 | Baseline Ornstein–Uhlenbeck; `initial_cond` = $[0.0]$, `theta` = 0.2, `mu` = 0.0, `sigma` = 0.3, `method` = "euler". |
| OU | `OU_PARAM` | 0.5 | 25000 | Parameter shock (`shock_kind` = `"param"`): `mu` : $0.0 \rightarrow 0.5$; other OU parameters as in `OU_BASE`. |
| SLDS | `SLDS_BASE` | 0.01 | 25000 | Baseline switching linear dynamical system; `A1` = 0.9, `Q1` = 0.05, `A2` = 0.98, `Q2` = 0.35, `p11` = 0.94, `p22` = 0.95. |
| SLDS | `SLDS_PARAM` | 0.01 | 25000 | Parameter shock (`shock_kind` = `"param"`): `A1` : $0.9 \rightarrow 0.83$, `Q1` : $0.05 \rightarrow 0.50$, `A2` : $0.98 \rightarrow 0.97$, `Q2` : $0.35 \rightarrow 0.30$, `p11` : $0.94 \rightarrow 0.96$, `p22` : $0.95 \rightarrow 0.92$. |
| SLDS | `SLDS_SWITCH` | 0.01 | 25000 | Switch shock (`shock_kind` = `"switch"`): `switch_update` sets `A1` = 0.87, `Q1` = 0.07, `A2` = 0.99, `Q2` = 0.45, `p11` = 0.90, `p22` = 0.95. |
| Double-well | `DOUBLEWELL_BASE` | 0.5 | 25000 | Baseline double-well SDE (Euler API, step size 0.5); `a` = 1.5, `sigma` = 0.25, `seed` = 1955. |
| Double-well | `DOUBLEWELL_PARAM` | 0.5 | 25000 | Parameter shock (`shock_kind` = `"param"`): `a` : $1.5 \rightarrow 1.0$, `sigma` : $0.25 \rightarrow 0.35$. |
| Double-well | `DOUBLEWELL_SWITCH` | 0.5 | 25000 | Switch shock (`shock_kind` = `"switch"`): `switch_update` sets `a` = 1.0, `sigma` = 0.35 (same target as `DOUBLEWELL_PARAM`). |
| Seasonal AR | `SEASONAL_AR_BASE` | 0.01 | 25000 | Baseline seasonal AR process (discrete-time; `dt`/`method` kept for API): `S` = 24, `phi` = 0.5, `sigma` = 0.2, `a0` = 1.0, `amp_drift_per_step` = 0. |
| Seasonal AR | `SEASONAL_AR_PARAM` | 0.01 | 25000 | Parameter shock (`shock_kind` = `"param"`): `a0` : $1.0 \rightarrow 1.4$, `sigma` : $0.2 \rightarrow 0.35$, `phi` : $0.5 \rightarrow 0.8$. |
| GARCH(1,1) | `GARCH_BASE` | 0.01 | 25000 | Baseline GARCH(1,1) volatility model (discrete-time; `dt`/`method` kept for API): `omega` = 0.01, `alpha` = 0.06, `beta` = 0.90. |
| GARCH(1,1) | `GARCH_PARAM` | 0.01 | 25000 | Parameter shock (`shock_kind` = `"param"`): `omega` : $0.01 \rightarrow 0.03$, `alpha` : $0.06 \rightarrow 0.15$, `beta` : $0.90 \rightarrow 0.70$. |
| KS | `KS_BASE` | 0.01 | 25000 | Baseline Kuramoto–Sivashinsky; `nx` = 64, `Lx` = 22.0, `nu` = 1.0, `method` = "etdrk4". |
| KS | `KS_PARAM` | 0.01 | 25000 | Parameter shock (`shock_kind` = `"param"`): `nu` : $1.0 \rightarrow 0.80$ (other KS parameters as in `KS_BASE`). |

Table 6: Parameter settings for the main chaotic benchmarks (top block) and all synthetic shock scenarios used in our experiments (bottom block). Shock scenarios are instantiated via `PremadeID.*` in `make_source`, with `shock_frac` fixed to 0.35 so that shocks are applied after the first 35% of the trajectory.

# B APPENDIX: DETAIL TABLES

## B.1 ETTM2

Table 7: ETTm2. Values are mean $\pm$ s.e. across 4 seeds [7, 1955, 2023, 4]. Higher is better for EPT, lower is better for the rest. Weighted averages down-weighs easier portions (e.g. 96 horizon error has weight $96/(96 + 192 + 336 + 720)$)

| Data | Hor. | Model | MSE | MAE | SWD | EPT |
|---|---|---|---|---|---|---|
| ETTm2 | 96 | FERN | $8.22 \pm 0.50$ | $1.89 \pm 0.08$ | $5.80 \pm 0.67$ | $91.43 \pm 0.14$ |
| | | TimeMixer | $7.33 \pm 0.55$ | $1.86 \pm 0.03$ | $5.39 \pm 0.43$ | $92.68 \pm 1.92$ |
| | | PatchTST | $6.88 \pm 0.63$ | $1.77 \pm 0.08$ | $5.21 \pm 0.77$ | $91.57 \pm 0.00$ |
| | | DLinear | $6.53 \pm 0.32$ | $1.79 \pm 0.05$ | $4.49 \pm 0.22$ | $93.79 \pm 2.21$ |
| | 192 | FERN | $15.17 \pm 1.19$ | $2.56 \pm 0.09$ | $10.90 \pm 1.01$ | $162.61 \pm 6.99$ |
| | | TimeMixer | $15.80 \pm 2.04$ | $2.62 \pm 0.16$ | $12.23 \pm 1.70$ | $171.40 \pm 3.71$ |
| | | PatchTST | $18.08 \pm 0.34$ | $2.77 \pm 0.02$ | $14.34 \pm 0.53$ | $168.12 \pm 2.52$ |
| | | DLinear | $15.92 \pm 1.10$ | $2.61 \pm 0.09$ | $12.36 \pm 0.87$ | $166.72 \pm 2.82$ |
| | 336 | FERN | $15.51 \pm 1.80$ | $2.56 \pm 0.17$ | $11.32 \pm 1.79$ | $285.76 \pm 0.57$ |
| | | TimeMixer | $24.16 \pm 1.99$ | $3.11 \pm 0.16$ | $19.36 \pm 2.07$ | $278.57 \pm 6.47$ |
| | | PatchTST | $22.97 \pm 2.46$ | $3.04 \pm 0.16$ | $18.73 \pm 2.20$ | $283.44 \pm 1.86$ |
| | | DLinear | $23.51 \pm 1.91$ | $3.06 \pm 0.13$ | $19.02 \pm 2.00$ | $273.09 \pm 9.58$ |
| | 720 | FERN | $15.36 \pm 2.71$ | $2.69 \pm 0.21$ | $8.31 \pm 2.67$ | $388.07 \pm 24.93$ |
| | | TimeMixer | $12.87 \pm 0.96$ | $2.40 \pm 0.10$ | $4.83 \pm 0.58$ | $397.42 \pm 40.43$ |
| | | PatchTST | $14.58 \pm 1.51$ | $2.56 \pm 0.10$ | $7.27 \pm 0.53$ | $437.14 \pm 47.60$ |
| | | DLinear | $15.99 \pm 1.17$ | $2.75 \pm 0.11$ | $8.03 \pm 1.31$ | $430.62 \pm 25.38$ |
| *Simple Average* | | FERN | 13.5650 | 2.4250 | 9.0825 | 231.9675 |
| | | TimeMixer | 15.04 | 2.50 | 10.45 | 235.02 |
| | | PatchTST | 15.63 | 2.54 | 11.39 | 245.07 |
| | | DLinear | 15.49 | 2.55 | 10.98 | 241.05 |
| *Weighted Average* | | FERN | 14.86 | 2.58 | 9.25 | 309.10 |
| | | TimeMixer | 15.72 | 2.57 | 9.56 | 313.65 |
| | | PatchTST | 16.63 | 2.65 | 11.00 | 335.60 |
| | | DLinear | 17.18 | 2.74 | 11.14 | 329.48 |

## B.2 CHUA'S CIRCUIT

Table 8: Chua's circuit, dt=5e-3, step=35,000. Values are mean $\pm$ s.e. across 4 seeds [7, 1955, 2023, 4]. Higher is better for EPT, lower is better for the rest. Weighted averages down-weighs easier portions (e.g. 96 horizon error has weight $96/(96 + 192 + 336 + 720)$)

| Data | Hor. | Model | MSE | MAE | SWD | EPT |
|------|------|-------|-----|-----|-----|-----|
| Chua | 96 | FERN | $0.0007 \pm 0.00$ | $0.0191 \pm 0.01$ | $0.0007 \pm 0.00$ | $96.0000 \pm 0.00$ |
| | | TimeMixer | $0.0015 \pm 0.00$ | $0.0259 \pm 0.00$ | $0.0014 \pm 0.00$ | $96.0000 \pm 0.00$ |
| | | PatchTST | $0.0063 \pm 0.00$ | $0.0509 \pm 0.01$ | $0.0057 \pm 0.00$ | $96.0000 \pm 0.00$ |
| | | DLinear | $0.0168 \pm 0.00$ | $0.0796 \pm 0.01$ | $0.0167 \pm 0.00$ | $96.0000 \pm 0.00$ |
| | 192 | FERN | $0.0011 \pm 0.00$ | $0.0257 \pm 0.00$ | $0.0009 \pm 0.00$ | $192.0000 \pm 0.00$ |
| | | TimeMixer | $0.0349 \pm 0.01$ | $0.1199 \pm 0.01$ | $0.0319 \pm 0.01$ | $192.0000 \pm 0.00$ |
| | | PatchTST | $0.1242 \pm 0.06$ | $0.1924 \pm 0.05$ | $0.1091 \pm 0.05$ | $192.0000 \pm 0.00$ |
| | | DLinear | $0.0265 \pm 0.01$ | $0.1183 \pm 0.02$ | $0.0246 \pm 0.01$ | $192.0000 \pm 0.00$ |
| | 336 | FERN | $0.0010 \pm 0.00$ | $0.0225 \pm 0.00$ | $0.0004 \pm 0.00$ | $336.0000 \pm 0.00$ |
| | | TimeMixer | $0.0045 \pm 0.00$ | $0.0479 \pm 0.01$ | $0.0028 \pm 0.00$ | $336.0000 \pm 0.00$ |
| | | PatchTST | $0.0278 \pm 0.01$ | $0.1060 \pm 0.02$ | $0.0215 \pm 0.01$ | $336.0000 \pm 0.00$ |
| | | DLinear | $0.1590 \pm 0.01$ | $0.2703 \pm 0.01$ | $0.0813 \pm 0.01$ | $336.0000 \pm 0.00$ |
| | 720 | FERN | $0.3301 \pm 0.51$ | $0.2386 \pm 0.17$ | $0.1994 \pm 0.33$ | $473.6345 \pm 72.58$ |
| | | TimeMixer | $0.7624 \pm 0.82$ | $0.4475 \pm 0.26$ | $0.5685 \pm 0.64$ | $426.9357 \pm 80.40$ |
| | | PatchTST | $1.7924 \pm 0.52$ | $0.9209 \pm 0.11$ | $1.3499 \pm 0.43$ | $228.1272 \pm 45.00$ |
| | | DLinear | $1.3719 \pm 0.09$ | $0.7314 \pm 0.03$ | $0.8305 \pm 0.07$ | $118.5541 \pm 2.00$ |
| *Simple Average* | | FERN | 0.0832 | 0.0765 | 0.0504 | 274.4086 |
| | | TimeMixer | 0.20083 | 0.16030 | 0.15115 | 262.73 |
| | | PatchTST | 0.48767 | 0.31755 | 0.37155 | 213.03 |
| | | DLinear | 0.39355 | 0.29990 | 0.23828 | 185.64 |
| *Weighted Average* | | FERN | 0.1773 | 0.1385 | 0.1071 | 372.0185 |
| | | TimeMixer | 0.41465 | 0.27069 | 0.30991 | 347.00 |
| | | PatchTST | 0.98536 | 0.55096 | 0.74453 | 240.50 |
| | | DLinear | 0.77968 | 0.48198 | 0.46994 | 181.80 |

## B.3 ROSSLER

Table 9: Rossler, dt=1e-2, step=25,000. Values are mean $\pm$ s.e. across 4 seeds $[7, 1955, 2023, 4]$. Lower is better for MSE/MAE/SWD; **higher is better for EPT**. Horizon-weighted averages means e.g. 96 horizon error has weight $96/(96 + 192 + 336 + 720)$, down-weighting easier portions of the forecast.

| Data | Hor. | Model | MSE | MAE | SWD | EPT |
|---|---|---|---|---|---|---|
| Rossler | 96 | FERN | $0.0032 \pm 0.00$ | $0.0445 \pm 0.01$ | $0.0030 \pm 0.00$ | $96.0000 \pm 0.00$ |
| | | TimeMixer | $0.1033 \pm 0.07$ | $0.2454 \pm 0.09$ | $0.0963 \pm 0.07$ | $96.0000 \pm 0.00$ |
| | | PatchTST | $0.1846 \pm 0.10$ | $0.3155 \pm 0.09$ | $0.1700 \pm 0.10$ | $96.0000 \pm 0.00$ |
| | | DLinear | $0.3036 \pm 0.09$ | $0.3343 \pm 0.06$ | $0.2908 \pm 0.08$ | $96.0000 \pm 0.00$ |
| | 192 | FERN | $0.0066 \pm 0.00$ | $0.0618 \pm 0.01$ | $0.0055 \pm 0.00$ | $192.0000 \pm 0.00$ |
| | | TimeMixer | $0.1124 \pm 0.03$ | $0.2689 \pm 0.03$ | $0.1016 \pm 0.03$ | $192.0000 \pm 0.00$ |
| | | PatchTST | $0.5704 \pm 0.14$ | $0.5893 \pm 0.09$ | $0.4762 \pm 0.14$ | $192.0000 \pm 0.00$ |
| | | DLinear | $3.6638 \pm 1.14$ | $1.0719 \pm 0.19$ | $3.2148 \pm 1.11$ | $179.2690 \pm 8.01$ |
| | 336 | FERN | $0.0830 \pm 0.03$ | $0.1988 \pm 0.04$ | $0.0567 \pm 0.02$ | $331.9004 \pm 4.31$ |
| | | TimeMixer | $14.9881 \pm 0.84$ | $2.4916 \pm 0.09$ | $12.8596 \pm 0.74$ | $162.4817 \pm 3.49$ |
| | | PatchTST | $23.9804 \pm 1.43$ | $3.3559 \pm 0.12$ | $20.8028 \pm 1.51$ | $69.0610 \pm 1.09$ |
| | | DLinear | $34.3263 \pm 1.34$ | $4.3221 \pm 0.09$ | $29.8861 \pm 1.22$ | $59.8618 \pm 1.16$ |
| | 720 | FERN | $0.0548 \pm 0.04$ | $0.1258 \pm 0.03$ | $0.0160 \pm 0.01$ | $676.7093 \pm 44.12$ |
| | | TimeMixer | $8.8210 \pm 0.60$ | $1.3566 \pm 0.06$ | $7.7532 \pm 0.47$ | $556.5020 \pm 21.15$ |
| | | PatchTST | $8.5990 \pm 0.19$ | $1.4545 \pm 0.02$ | $7.5545 \pm 0.16$ | $538.6870 \pm 3.03$ |
| | | DLinear | $8.2653 \pm 0.34$ | $1.5395 \pm 0.09$ | $7.3939 \pm 0.29$ | $594.3028 \pm 24.30$ |
| *Simple Average* | | FERN | 0.0369 | 0.1077 | 0.0203 | 324.1524 |
| | | TimeMixer | 6.0062 | 1.0906 | 5.2027 | 251.75 |
| | | PatchTST | 8.3336 | 1.4288 | 7.2509 | 223.94 |
| | | DLinear | 11.6398 | 1.8170 | 10.1964 | 232.36 |
| *Weighted Average* | | FERN | 0.0513 | 0.1291 | 0.0237 | 479.7837 |
| | | TimeMixer | 8.4960 | 1.4056 | 7.3898 | 373.03 |
| | | PatchTST | 10.6964 | 1.7249 | 9.3279 | 340.13 |
| | | DLinear | 13.5545 | 2.0823 | 11.9126 | 365.81 |

## B.4 ETTн2

Table 10: ETTh2. Values are mean $\pm$ s.e. across 4 seeds $[7, 1955, 2023, 4]$. Lower is better for MSE/MAE/SWD; **higher is better for EPT**. horizon-weighted averages means e.g. 96 horizon error has weight $96/(96 + 192 + 336 + 720)$, down-weighting easier portions of the forecast.

| Data | Hor. | Model | MSE | MAE | SWD | EPT |
|------|------|-------|-----|-----|-----|-----|
| ETTh2 | 96 | FERN | $11.71 \pm 1.46$ | $2.28 \pm 0.15$ | $6.03 \pm 1.16$ | $79.73 \pm 2.26$ |
| | | TimeMixer | $8.02 \pm 0.39$ | $1.81 \pm 0.04$ | $3.52 \pm 0.25$ | $86.00 \pm 0.66$ |
| | | PatchTST | $8.22 \pm 0.35$ | $1.84 \pm 0.03$ | $3.95 \pm 0.15$ | $85.67 \pm 0.73$ |
| | | DLinear | $8.74 \pm 0.37$ | $1.88 \pm 0.03$ | $3.61 \pm 0.33$ | $85.29 \pm 0.29$ |
| | 192 | FERN | $11.19 \pm 0.07$ | $2.32 \pm 0.02$ | $4.74 \pm 0.07$ | $140.40 \pm 4.66$ |
| | | TimeMixer | $12.31 \pm 0.51$ | $2.35 \pm 0.02$ | $4.99 \pm 0.59$ | $133.96 \pm 4.62$ |
| | | PatchTST | $11.91 \pm 0.78$ | $2.29 \pm 0.07$ | $4.32 \pm 0.59$ | $142.60 \pm 2.91$ |
| | | DLinear | $11.49 \pm 0.72$ | $2.26 \pm 0.06$ | $4.09 \pm 0.10$ | $142.62 \pm 3.97$ |
| | 336 | FERN | $29.80 \pm 1.53$ | $3.78 \pm 0.07$ | $25.08 \pm 2.02$ | $130.53 \pm 3.06$ |
| | | TimeMixer | $19.80 \pm 2.40$ | $3.07 \pm 0.18$ | $15.03 \pm 2.36$ | $176.68 \pm 7.23$ |
| | | PatchTST | $18.44 \pm 1.76$ | $3.02 \pm 0.17$ | $13.27 \pm 1.71$ | $165.09 \pm 3.69$ |
| | | DLinear | $18.41 \pm 1.44$ | $2.98 \pm 0.13$ | $12.94 \pm 1.15$ | $172.96 \pm 9.70$ |
| | 720 | FERN | $16.95 \pm 2.12$ | $2.91 \pm 0.16$ | $8.64 \pm 1.53$ | $198.81 \pm 12.41$ |
| | | TimeMixer | $16.67 \pm 2.16$ | $2.61 \pm 0.17$ | $10.61 \pm 2.32$ | $289.13 \pm 5.03$ |
| | | PatchTST | $19.20 \pm 1.79$ | $2.85 \pm 0.15$ | $12.35 \pm 1.12$ | $284.39 \pm 13.47$ |
| | | DLinear | $19.26 \pm 1.73$ | $2.86 \pm 0.13$ | $11.04 \pm 1.16$ | $335.81 \pm 5.44$ |
| *Simple Average* | | FERN | 17.41 | 2.82 | 11.12 | 137.3675 |
| | | TimeMixer | 14.20 | 2.46 | 8.54 | 171.44 |
| | | PatchTST | 14.44 | 2.50 | 8.47 | 169.44 |
| | | DLinear | 14.48 | 2.49 | 7.92 | 184.17 |
| *Weighted Average* | | FERN | 18.97 | 30.. | 12.00 | 164.89 |
| | | TimeMixer | 16.21 | 2.63 | 10.41 | 224.34 |
| | | PatchTST | 17.18 | 2.74 | 10.83 | 220.12 |
| | | DLinear | 17.19 | 2.73 | 9.99 | 249.60 |

## B.5 ETTH1

Table 11: ETTh1. Values are mean ± s.e. across 4 seeds [7, 1955, 2023, 4]. Lower is better for MSE/MAE/SWD; **higher is better for EPT**. horizon-weighted averages means e.g. 96 horizon error has weight $96/(96 + 192 + 336 + 720)$, down-weighting easier portions of the forecast.

| Data | Hor. | Model | MSE | MAE | SWD | EPT |
|------|------|-------|-----|-----|-----|-----|
| ETTh1 | 96 | FERN | $6.68 \pm 0.05$ | $1.50 \pm 0.01$ | $2.88 \pm 0.09$ | $34.11 \pm 0.68$ |
| | | TimeMixer | $6.85 \pm 0.43$ | $1.47 \pm 0.04$ | $3.02 \pm 0.28$ | $39.40 \pm 1.31$ |
| | | PatchTST | $5.87 \pm 0.14$ | $1.43 \pm 0.03$ | $2.48 \pm 0.08$ | $41.45 \pm 1.15$ |
| | | DLinear | $6.62 \pm 0.06$ | $1.45 \pm 0.01$ | $2.77 \pm 0.10$ | $38.53 \pm 1.15$ |
| | 192 | FERN | $6.21 \pm 0.19$ | $1.49 \pm 0.03$ | $1.81 \pm 0.13$ | $61.29 \pm 7.42$ |
| | | TimeMixer | $6.82 \pm 0.08$ | $1.49 \pm 0.02$ | $2.21 \pm 0.10$ | $70.84 \pm 4.58$ |
| | | PatchTST | $6.12 \pm 0.24$ | $1.48 \pm 0.03$ | $1.81 \pm 0.11$ | $78.49 \pm 8.21$ |
| | | DLinear | $7.39 \pm 0.17$ | $1.51 \pm 0.03$ | $2.36 \pm 0.17$ | $81.34 \pm 6.13$ |
| | 336 | FERN | $6.41 \pm 0.33$ | $1.54 \pm 0.03$ | $3.25 \pm 0.34$ | $68.77 \pm 7.36$ |
| | | TimeMixer | $6.06 \pm 0.20$ | $1.43 \pm 0.01$ | $3.27 \pm 0.28$ | $72.88 \pm 1.20$ |
| | | PatchTST | $6.81 \pm 0.47$ | $1.56 \pm 0.05$ | $3.75 \pm 0.63$ | $69.49 \pm 1.49$ |
| | | DLinear | $6.39 \pm 0.05$ | $1.48 \pm 0.01$ | $3.16 \pm 0.11$ | $74.75 \pm 2.14$ |
| | 720 | FERN | $7.09 \pm 0.20$ | $1.57 \pm 0.06$ | $2.60 \pm 0.19$ | $112.16 \pm 18.48$ |
| | | TimeMixer | $7.57 \pm 0.45$ | $1.69 \pm 0.07$ | $2.80 \pm 0.35$ | $125.88 \pm 2.06$ |
| | | PatchTST | $7.67 \pm 0.14$ | $1.70 \pm 0.05$ | $3.03 \pm 0.15$ | $121.38 \pm 1.47$ |
| | | DLinear | $7.77 \pm 0.15$ | $1.67 \pm 0.05$ | $2.70 \pm 0.11$ | $124.28 \pm 0.78$ |
| *Simple Average* | | FERN | 6.60 | 1.53 | 2.64 | 69.08 |
| | | TimeMixer | 6.83 | 1.52 | 2.83 | 77.25 |
| | | PatchTST | 6.62 | 1.54 | 2.77 | 77.70 |
| | | DLinear | 7.04 | 1.53 | 2.75 | 79.72 |
| *Weighted Average* | | FERN | 6.77 | 1.55 | 2.67 | 88.47 |
| | | TimeMixer | 7.03 | 1.58 | 2.85 | 98.59 |
| | | PatchTST | 7.10 | 1.61 | 3.00 | 96.57 |
| | | DLinear | 7.29 | 1.58 | 2.77 | 99.64 |

## B.6 ETTM1

Table 12: ETTm1. Values are mean $\pm$ s.e. across 4 seeds $[7, 1955, 2023, 4]$. Lower is better for MSE/MAE/SWD; **higher is better for EPT**. Horizon-weighted averages use weights $96, 192, 336, 720$ (i.e., 96 has weight $96/(96+192+336+720)$).

| Data | Hor. | Model | MSE | MAE | SWD | EPT |
|------|------|-------|-----|-----|-----|-----|
| ETTm1 | 96 | FERN | $2.67 \pm 0.03$ | $1.07 \pm 0.03$ | $0.95 \pm 0.08$ | $44.32 \pm 1.27$ |
| | | TimeMixer | $2.91 \pm 0.64$ | $0.98 \pm 0.09$ | $1.07 \pm 0.45$ | $46.79 \pm 2.80$ |
| | | PatchTST | $3.03 \pm 0.36$ | $1.04 \pm 0.06$ | $1.14 \pm 0.31$ | $43.57 \pm 0.35$ |
| | | DLinear | $2.33 \pm 0.11$ | $0.91 \pm 0.03$ | $1.01 \pm 0.11$ | $43.32 \pm 2.22$ |
| | 192 | FERN | $6.86 \pm 0.69$ | $1.59 \pm 0.06$ | $5.30 \pm 0.58$ | $66.75 \pm 1.64$ |
| | | TimeMixer | $5.77 \pm 0.48$ | $1.62 \pm 0.09$ | $4.32 \pm 0.38$ | $69.97 \pm 4.67$ |
| | | PatchTST | $6.60 \pm 1.60$ | $1.54 \pm 0.13$ | $4.96 \pm 1.45$ | $67.17 \pm 2.99$ |
| | | DLinear | $7.10 \pm 0.33$ | $1.50 \pm 0.04$ | $5.60 \pm 0.36$ | $69.99 \pm 1.90$ |
| | 336 | FERN | $6.42 \pm 0.56$ | $1.57 \pm 0.05$ | $3.45 \pm 0.72$ | $105.69 \pm 4.66$ |
| | | TimeMixer | $6.04 \pm 0.32$ | $1.55 \pm 0.08$ | $3.61 \pm 0.41$ | $107.72 \pm 3.85$ |
| | | PatchTST | $5.75 \pm 0.22$ | $1.45 \pm 0.06$ | $2.31 \pm 0.50$ | $101.94 \pm 4.42$ |
| | | DLinear | $7.98 \pm 0.23$ | $1.56 \pm 0.04$ | $5.37 \pm 0.28$ | $111.09 \pm 0.47$ |
| | 720 | FERN | $7.26 \pm 0.43$ | $1.58 \pm 0.05$ | $1.68 \pm 0.18$ | $162.74 \pm 16.46$ |
| | | TimeMixer | $6.35 \pm 0.15$ | $1.41 \pm 0.03$ | $1.39 \pm 0.28$ | $207.69 \pm 15.89$ |
| | | PatchTST | $6.06 \pm 0.14$ | $1.44 \pm 0.02$ | $1.35 \pm 0.22$ | $212.47 \pm 9.95$ |
| | | DLinear | $7.82 \pm 0.07$ | $1.58 \pm 0.00$ | $2.61 \pm 0.15$ | $186.79 \pm 13.50$ |
| *Simple Average* | | FERN | 5.80 | 1.45 | 2.85 | 94.88 |
| | | TimeMixer | 5.27 | 1.39 | 2.60 | 108.04 |
| | | PatchTST | 5.36 | 1.37 | 2.44 | 106.29 |
| | | DLinear | 6.31 | 1.39 | 3.65 | 102.80 |
| *Weighted Average* | | FERN | 6.67 | 1.54 | 2.59 | 126.31 |
| | | TimeMixer | 5.94 | 1.44 | 2.34 | 151.53 |
| | | PatchTST | 5.84 | 1.43 | 2.09 | 152.02 |
| | | DLinear | 7.37 | 1.52 | 3.61 | 140.93 |

## B.7 WEATHER

Table 13: Weather with normalized scale. Values are mean $\pm$ s.e. across 4 seeds $[7, 1955, 2023, 4]$. Lower is better for MSE/MAE/SWD; **higher is better for EPT**. horizon-weighted averages means e.g. 96 horizon error has weight $96/(96 + 192 + 336 + 720)$, down-weighting easier portions of the forecast.

| Data | Hor. | Model | MSE | MAE | SWD | EPT |
|------|------|-------|-----|-----|-----|-----|
| Weather | 96 | FERN | $0.10 \pm 0.01$ | $0.21 \pm 0.02$ | $0.07 \pm 0.01$ | $85.07 \pm 0.94$ |
| | | TimeMixer | $0.09 \pm 0.01$ | $0.17 \pm 0.01$ | $0.07 \pm 0.01$ | $85.23 \pm 0.46$ |
| | | PatchTST | $0.06 \pm 0.00$ | $0.16 \pm 0.00$ | $0.04 \pm 0.00$ | $86.08 \pm 0.17$ |
| | | DLinear | $0.06 \pm 0.00$ | $0.16 \pm 0.01$ | $0.04 \pm 0.00$ | $85.88 \pm 0.11$ |
| | 192 | FERN | $0.12 \pm 0.02$ | $0.25 \pm 0.02$ | $0.08 \pm 0.02$ | $165.61 \pm 2.21$ |
| | | TimeMixer | $0.11 \pm 0.02$ | $0.20 \pm 0.02$ | $0.07 \pm 0.01$ | $162.69 \pm 1.07$ |
| | | PatchTST | $0.11 \pm 0.01$ | $0.20 \pm 0.01$ | $0.08 \pm 0.01$ | $164.53 \pm 2.10$ |
| | | DLinear | $0.08 \pm 0.00$ | $0.21 \pm 0.01$ | $0.05 \pm 0.00$ | $167.43 \pm 0.40$ |
| | 336 | FERN | $0.19 \pm 0.04$ | $0.28 \pm 0.03$ | $0.14 \pm 0.04$ | $268.22 \pm 2.88$ |
| | | TimeMixer | $0.30 \pm 0.14$ | $0.29 \pm 0.05$ | $0.23 \pm 0.11$ | $269.45 \pm 1.63$ |
| | | PatchTST | $0.21 \pm 0.02$ | $0.26 \pm 0.01$ | $0.16 \pm 0.02$ | $272.48 \pm 2.53$ |
| | | DLinear | $0.10 \pm 0.00$ | $0.21 \pm 0.00$ | $0.07 \pm 0.00$ | $282.22 \pm 0.92$ |
| | 720 | FERN | $0.68 \pm 0.05$ | $0.44 \pm 0.02$ | $0.58 \pm 0.03$ | $472.59 \pm 11.00$ |
| | | TimeMixer | $0.58 \pm 0.02$ | $0.36 \pm 0.01$ | $0.50 \pm 0.02$ | $456.41 \pm 2.20$ |
| | | PatchTST | $0.59 \pm 0.03$ | $0.37 \pm 0.01$ | $0.50 \pm 0.03$ | $451.95 \pm 2.58$ |
| | | DLinear | $0.58 \pm 0.00$ | $0.39 \pm 0.00$ | $0.50 \pm 0.01$ | $469.59 \pm 7.63$ |
| *Simple Average* | | FERN | 0.27 | 0.30 | 0.22 | 247.87 |
| | | TimeMixer | 0.27 | 0.26 | 0.22 | 243.44 |
| | | PatchTST | 0.24 | 0.25 | 0.20 | 243.76 |
| | | DLinear | 0.21 | 0.24 | 0.16 | 251.28 |
| *Weighted Average* | | FERN | 0.44 | 0.36 | 0.36 | 349.96 |
| | | TimeMixer | 0.41 | 0.31 | 0.34 | 341.20 |
| | | PatchTST | 0.39 | 0.30 | 0.32 | 339.89 |
| | | DLinear | 0.35 | 0.30 | 0.29 | 352.17 |

## B.8 LORENZ

Table 14: Lorenz. Values are mean $\pm$ s.e. across 4 seeds $[7, 1955, 2023, 4]$. Lower is better for MSE/MAE/SWD; **higher is better for EPT**. horizon-weighted averages means e.g. 96 horizon error has weight $96/(96 + 192 + 336 + 720)$, down-weighting easier portions of the forecast.

| Data | Hor. | Model | MSE | MAE | SWD | EPT |
|------|------|-------|-----|-----|-----|-----|
| Lorenz | 96 | FERN | $0.47 \pm 0.15$ | $0.50 \pm 0.08$ | $0.21 \pm 0.06$ | $96.00 \pm 0.00$ |
| | | TimeMixer | $0.18 \pm 0.01$ | $0.33 \pm 0.01$ | $0.10 \pm 0.01$ | $96.00 \pm 0.00$ |
| | | PatchTST | $1.17 \pm 0.31$ | $0.83 \pm 0.09$ | $0.86 \pm 0.22$ | $96.00 \pm 0.00$ |
| | | DLinear | $54.04 \pm 3.03$ | $4.92 \pm 0.17$ | $33.48 \pm 1.79$ | $50.42 \pm 2.88$ |
| | 192 | FERN | $2.06 \pm 0.69$ | $0.83 \pm 0.13$ | $0.33 \pm 0.09$ | $175.23 \pm 6.90$ |
| | | TimeMixer | $16.91 \pm 16.46$ | $2.45 \pm 1.20$ | $8.55 \pm 8.93$ | $136.16 \pm 56.75$ |
| | | PatchTST | $6.90 \pm 2.06$ | $1.91 \pm 0.29$ | $2.52 \pm 0.80$ | $179.67 \pm 12.43$ |
| | | DLinear | $79.25 \pm 2.61$ | $6.78 \pm 0.15$ | $48.51 \pm 1.24$ | $19.76 \pm 0.18$ |
| | 336 | FERN | $21.25 \pm 8.18$ | $2.39 \pm 0.41$ | $3.49 \pm 1.38$ | $187.03 \pm 12.57$ |
| | | TimeMixer | $55.97 \pm 2.09$ | $5.10 \pm 0.07$ | $25.41 \pm 1.99$ | $92.78 \pm 7.91$ |
| | | PatchTST | $60.96 \pm 9.40$ | $5.43 \pm 0.44$ | $25.12 \pm 2.81$ | $105.74 \pm 14.05$ |
| | | DLinear | $61.06 \pm 1.31$ | $5.64 \pm 0.12$ | $30.28 \pm 0.37$ | $79.80 \pm 1.02$ |
| | 720 | FERN | $63.52 \pm 2.25$ | $4.96 \pm 0.20$ | $4.89 \pm 0.57$ | $293.46 \pm 14.90$ |
| | | TimeMixer | $50.71 \pm 15.19$ | $4.91 \pm 0.40$ | $10.39 \pm 3.56$ | $247.14 \pm 13.47$ |
| | | PatchTST | $51.41 \pm 6.62$ | $4.94 \pm 0.24$ | $9.89 \pm 2.96$ | $254.76 \pm 29.58$ |
| | | DLinear | $76.69 \pm 1.48$ | $6.96 \pm 0.11$ | $40.59 \pm 0.29$ | $24.01 \pm 3.50$ |
| *Simple Average* | | FERN | 21.82 | 2.17 | 2.23 | 187.93 |
| | | TimeMixer | 30.94 | 3.19 | 11.11 | 143.02 |
| | | PatchTST | 30.11 | 3.28 | 9.60 | 159.04 |
| | | DLinear | 67.76 | 6.07 | 38.22 | 43.50 |
| *Weighted Average* | | FERN | 39.67 | 3.41 | 3.55 | 235.85 |
| | | TimeMixer | 43.58 | 4.28 | 13.15 | 181.90 |
| | | PatchTST | 43.85 | 4.33 | 12.00 | 195.44 |
| | | DLinear | 71.53 | 6.46 | 38.64 | 39.24 |

## C ARCHITECTURE

All timings in Table 16 were obtained on a single NVIDIA RTX 4070 (8 GB VRAM). FLOPs are reported per forward pass using THOP (Zhu, 2019). Exact software versions and runtime scripts are included in the code release.

Table 15: FERN architecture and training hyperparameters.

| Parameter | Value |
|---|---|
| *General Training Hyperparameters* | |
| Optimizer | AdamW, no weight decay |
| Learning Rate | $3 \times 10^{-4}$ |
| Batch Size | 96 |
| Layer Norm & Dropout | No |
| Input Patch Length & Lookback Window | 24 & 336 |
| Epochs, Patience, Seeds | 50, 5, [20, 7, 1955, 2023, 4] |
| *Component-wise Architecture* | |
| Activation Function | ELU + LogSigmiod; |
| Latent Dimension ($d_z$) | 144 |
| Hidden/Feature Dimension $d_h$ | 144 |
| Shared MLP Hidden Dimension | [e.g., 256] |
| Encoder & Decoder Stacks | 6 & 4 |
| Shared MLP Architecture | 11 layers (incl. 5 acti.) |
| Affine Head Architecture | 5 layers (incl. 3 acti.) |
| Householder Head Architecture | 2 layers (incl. 1 acti.) |
| Householder Reflections | 8 |

Table 16: Compute footprint on 52k steps Lorenz63 (336-in-336-out) and ETTm1 (96-in-336-out).

| Model | Training time | | Params | GFLOPs, per sample, per step |
|---|---|---|---|---|
| | Lorenz (min) | ETTm1 (s) | (M) | (G) |
| FERN | 16.0 | 83 | 1.025 | 0.0035 |
| TimeMixer | 21.0 | 120 | 0.886 | 0.0463 |
| PatchTST | 10.0 | 110 | 2.008 | 0.0308 |
| DLinear | 2.5 | 27 | 0.679 | 0.0007 |

