# OpenReview forum: "Chaining Spectral Pearls: Ellipsoidal Forecasting Beyond Trajectories for Time Series"
_ICLR.cc/2026/Conference — Submitted to ICLR 2026_

### Official Review · Reviewer_ZSmM · 2025-10-27

**Soundness:** 2
**Presentation:** 1
**Contribution:** 2
**Rating:** 2
**Confidence:** 4

**Summary:**

The paper introduces FERN (Forecasting with Ellipsoidal Rep-resentatioN), a time-series forecaster that predicts future geometry (ellipsoids) rather than exact trajectories, which is robust under deterministic chaos. It employs local linear transport with explicit spectral factors (eigenvectors/eigenvalues) for interpretability. FERN is stress-tested on chaotic systems (Lorenz63, Rössler, Chua) using new metrics like Sliced Wasserstein Distance and Effective Prediction Time, and outperforms baselines significantly on these.

**Strengths:**

1. The paper proposes new metrics (Sliced Wasserstein Distance and Effective Prediction Time) for evaluating forecasting performance in chaotic systems.
2. The evaluation involving real-world benchmarks enhances the practical relevance of the proposed method.

**Weaknesses:**

1. The Appendix is unstructured and requires more rigorous revision to meet the publication standard.
2. The literature review is insufficient, and many related works and baselines are not discussed, making it difficult to assess the novelty and effectiveness of the proposed method. For instances:
 - Koopman operator-based methods for modeling chaos: Cheng, Xiaoyuan, et al. "Learning Chaos In A Linear Way." The Thirteenth International Conference on Learning Representations.
 - Geometric distribution preserving methods: Li, Zongyi, et al. "Learning Dissipative Dynamics in Chaotic Systems." (NeurIPS 2021).
 - Chaos system Benchmarks: Gilpin, William. "Chaos as an interpretable benchmark for forecasting and data-driven modelling." Thirty-fifth Conference on Neural Information Processing Systems Datasets and Benchmarks Track (Round 2).
 - Reservoir computing methods for chaotic systems.
 - Diffusion/Flow based methods, e.g. Shysheya, Aliaksandra, et al. "On conditional diffusion models for PDE simulations." Advances in Neural Information Processing Systems 37 (2024): 23246-23300.

3. The paper lacks clarity in validating the method in how is the spectral factors in deployed; how the network is implemented and optimized with the Algorithm 1.

**Questions:**

1. How is future geometry defined formally?
2. How is the model's performance compared to existing methods for chaotic time series forecasting as referenced above?
3. Interest in scalability of Rotation: for high-dimensional data, how does the fixed R=8 reflections sufficiently approximate the necessary rotation U(z), and is this rotation layer a bottleneck for datasets larger than the 21 variables in Weather?

---

> ### Author Response · Authors · 2025-11-20
> **Response to Reviewer ZSmM: Clarifying Local Geometry, and Scalability**
>
> We sincerely thank the reviewer for his suggestions of relevant literature and extensive comments, and would like to give a preliminary response to address the theoretical questions.
>
> ### On "Appendix Lacking Structure"
> The reviewer is **absolutely right**.
>
> **Action taken**: We acknowledge the problem and will fix the format in the upcoming new version.
>
> ## On "Local Geometry and Scalability"
> > How is future geometry defined formally.
>
> > Interest in scalability of Rotation: for high-dimensional data, how does the fixed R=8 reflections sufficiently approximate the necessary rotation U(z), and is this rotation layer a bottleneck for datasets larger than the 21 variables in Weather?
>
> We try to provide a succinct answer from a scalability angle.
>
> &nbsp;
>
> **Traditional Dynamics**: Learn a vector field map F "rolling" *data* over time, **autoregressively**: we hope $F(y_{0:t}) \approx y_{t:2t}$ and $F(y_{t:2t}) \approx y_{2t:3t}$.
>
> &nbsp;
>
> **By Brenier's Theorem**, from abs.cont source such as t-dim N(0,I), **there exists a unique** OT from N(0,I) to true target y_{t:2t}, or any distribution (with finite second moments) produced by F. Similarly, when **we divided length-t dim into patches** with patch size p, the OT's from p-dim N(0, I) to segments {0:p}, {p:2p} ... also **uniquely exist**.
>
> &nbsp;
>
> **Our Approach**: Learn **parallel** G "pushing" *N(0,I)* to true target, mimicking the unique SPSD solution by *approximating* SPSD Jacobian A. **NOTE:** The Gaussian shape N(μ, A A^T) induced by A acting on N(μ,I) is the **conditional local geometry** of that patch.
>
> &nbsp;
>
> In short, **Dynamics vs Local geometry** is about using route F or route G:
>
> &nbsp;
>
> -F: formula is global and fixed across time, interpretable, hard to find, susceptible to nonstationarity. **Jacobian of F** is general: n-dim Jacobian has O(n^2) entries to search, expensive for **fully data-dependent** modeling.
>
> -G: data-dependent, geometrically interpretable, easier to find (SPSD is a much smaller function class), stable to nonstationarity, see new experiments in upcoming thread as proof.
>
> &nbsp;
>
> **The Preliminary Benefit**: *SPSD Jacobians* can be decomposed into $U \Lambda U^T$. **Fact: any n-dim orthogonal matrix U can be decomposed to *at most* n Householder products**. We can search low-rank, *fully* data-dependent Jacobian matrix with O(Rn) cost, where R=# of reflections. Eigenvalues and eigenvectors (for each patch) come "for free", without an O(n^3) decomposition cost. **NOTE:** the eigenvalues are not conflated by system eigenvalues: it is how *FERN* believes N(0,I) should stretch at that segment, so they are comparable across patches and rounds. See the new plots in an upcoming thread.
>
> &nbsp;
>
> **The Scalability Benefit**: prediction horizon=336, reflection R=8, a *low-rank* Jacobian matrix costs O(8\*336). Full rank costs: O(336^2). **With patching**, 336=14 size-24 patches, *full-rank* patch Jacobian cost O(24\*24), **trainable in parallel**. For dynamics F, we need to iteratively roll forward 14 times **sequentially**.
>
> &nbsp;
>
> **On Weather dataset**: the current implementation relies solely on Takens embedding theorem, so it is effectively *channel independent*. In the case of channel/feature explosion, we can treat each variable as a single sample.
>
> &nbsp;
>
> **On R=8 sufficiency**: we run an ablation test today to see the differences on Lorenz63, 25k samples. We implment an algorithm that updates one third reflections (detached the rest so no learning) that *decrease backprop memory cost*.
>
> &nbsp;
>
> Reflection: 8
> MSE=1.109 | MAE=0.570
> time: 484.86 seconds
>
> &nbsp;
>
> Reflection: 24 train with 3 size-8 blocks (updates one third each forward)
> MSE=1.286 | MAE=0.628
> time: 457.08 seconds
>
> &nbsp;
>
> Reflection: 24
> MSE=1.707 | MAE=0.662
> 471.50 seconds
>
> &nbsp;
>
> **Comment:** without patching, size=8 still works well in general. We conjecture it is because roughly as heuristic, **for a rotation in d dimension space, we need roughly d/2 reflections**; Lorenz is 3-dim and *genuinely* living a low manifold, so 2 reflections suffices (as authors we empirically knew that is true for the model).
>
> &nbsp;
>
> for ETT's, the actual dim may be low/medium, but the ETT's features suffer from data quality issues and likely provides overlapping information. Empirically, 2-8 suffice from our experience. We therefore fix R=8 across all datasets (including ETT) for simplicity and fairness.”
>
> &nbsp;
>
> A response to the literature critic will appear in a follow up thread.

---

> ### Author Response · Authors · 2025-11-21
> **Reply 2 to Reviewer ZSmM: Method Comparisons, New Experiments, and Spectral Plots**
>
> We posted results of **21 new experiments** on **non-stationarity** today, where FERN is shown to **not only dominate on chaotic datasets**, but also one of the **top performer in stochastic datasets** with different **controlled shocks**. Details are in posted new official comments. We invite the reviewer to look at those new experiments.
>
> &nbsp;
>
> We added four **new plots** to show the **advantage** of having *explicit eigenvalues*: **when the data changes rapidly, the local max eigenvalues increase rapidly and the timing of the responses **tracks** changes accurately.** It also shows where the model **fails**: on one side of the attractor, the *overall eigenvalue response (trace variation)* is more than necessary. Such diagnostics are available to users **despite an almost perfect reconstruction plot with very low MSE.** Details are in the posted official comments.
>
> &nbsp;
>
> ## On "Spectral Factors"
> > The paper lacks clarity in validating the method in how is the spectral factors in deployed; how the network is implemented and optimized with the Algorithm 1.
>
> The **four new plots** clearly demonstrate how the model's **spectral factors are deployed empirically.** On *network implementation*, we remove the Koopman theory section, which we recognize as a confusing and unhelpful detour. The streamlined model is simply:
>
> &nbsp;
>
> $U(z) \Lambda(z) U^T(z) x$ + t(z), where z is Gaussian latent.
>
> &nbsp;
>
> U(z) is a product of Householder reflections: **a 2-layer MLP generates unit-norm reflection vectors**, and we sequentially apply the reflections; The diagonal matrix $\Lambda(z)$ is just one vector s(z): **A 6-layer MLP generates parameters s(z), t(z)** to be multiplied and added in elementwise manner.
>
> &nbsp;
>
> **Action taken**: We include new plots, *carefully streamline the explanation and restructure appendix*, in the upcoming revision PDF.
>
> ## On "Literature Review and Methods Comparison"
> > The literature review is insufficient, and many related works and baselines are not discussed, .... For instances:...
>
> > How is the model's performance compared to existing methods for chaotic time series forecasting as referenced above?
>
> In the **21 new tests, we build a larger system benchmark including both stochastic datasets and chaotic datasets, with controlled non-stationarity**. As such, Gilpin's **static** benchmark, which is *exclusively for static chaotic datasets*, is interesting yet less relevant for LTSF.
>
> **Action taken**: **In the 21 new experiments we include model "PFNN" in [Cheng et al., 2025].** We  follow the GitHub and make sensible hyper-parameter choices. Results on plain Chaotic datasets (336-in-336-out):
>
> &nbsp;
>
> LORENZ_BASE (MSE ↓ / SWD ↓). Base = TimeMixer (tm).
>
> - DLinear  :  MSE 76.548, SWD 39.340  | 1.77× / 3.90× tm
> - Fern     :  MSE **21.663**, SWD **4.409**  | 0.50× / 0.44× tm
> - Koopa    :  MSE 95.501, SWD 11.049  | 2.21× / 1.10× tm
> - ModernTCN:  MSE *26.023*, SWD *5.962*  | 0.60× / 0.59× tm
> - PFNN     :  MSE 198.152, SWD 120.587  | 4.59× / 11.95× tm
> - TimeMixer:  MSE 43.209, SWD 10.087   | 1.00× / 1.00× tm
> - PatchTST :  MSE 38.889, SWD 10.563   | 0.90× / 1.05× tm
>
> &nbsp;
>
> ROSSLER_BASE.
>
> - DLinear  :  MSE 5.418, SWD 5.057    | 5.25× / 5.60× tm
> - Fern     :  MSE **0.019**, SWD **0.011**  | 0.02× / 0.01× tm
> - Koopa    :  MSE 11.941, SWD 5.579   | 11.57× / 6.18× tm
> - ModernTCN:  MSE *0.469*, SWD *0.421*    | 0.45× / 0.47× tm
> - PFNN     :  MSE 21.046, SWD 16.639  | 20.39× / 18.43× tm
> - TimeMixer:  MSE 1.032, SWD 0.903    | 1.00× / 1.00× tm
> - PatchTST :  MSE 2.447, SWD 2.248    | 2.37× / 2.49× tm
>
> **Further evidence**: we were concerned about the error numbers. Yet, [Cheng et al., 2025] reports NRMSE of 0.49 on 80k samples with dt=1e-2. Since Lorenz63 has standard deviation about 8, that translates to **roughly 15-20 MSE** on a 100 step rollout. Our result is about an **MSE of 1.2** on a 22k samples, 96 step prediction. While we **acknowledge** the reviewer's concern, We were **hesitant to make** a direct head-to-head comparison, as **they belong to different families**. This might create an **unfair impression**.
>
> &nbsp;
>
> **LTSF vs Markov Operator:** **LTSF e.g. predicts [Batch Feature Seq_{100:200}] based on [Batch Feature Seq_{0:100}]**, the Feature channel are untouched in most models(channel independent). **Markov Operator, predicts [Batch Seq_{1:2} Feature] based on [Batch Seq_{0:1} Feature]**, relying *solely* on the states information.
>
> &nbsp;
>
> Lorenz etc are **NOT** Markov systems, the past information informs the model about its likely current location. The task of *Markov Operator* is to *learn the global invariant measure* based on *current* state, which is a **strictly more difficult problem** than *learning conditional forecast distribution* with recent information. Lorenz63 has 3 state variables, **~10 variables are enough** to invoke Takens' Embedding Theorem, so the remaining input length serves as historical context.

---

> > ### Comment · Reviewer_ZSmM · 2025-11-22
> >
> > Thank you for the detailed replies and the additional results. I appreciate the efforts and am willing to raise my score to 4.  I am expecting the findings are clarified and merged to the **manuscript** in this stage to demonstrate the improvement, which ICLR exclusively encourages this.
> >
> > At the same time, many points raised in the authors’ response require clearer integration into the manuscript to ensure accuracy and coherence in acedemic positioning and writing. For example:
> >
> > 1. The rebuttal refers to a “Markov Operator,” but the correct notion in this literature is the Markov Neural Operator (MNO). This distinction is important, and I encourage the authors to adopt the established terminology before developing their arguments to keep the discussion in a professional way.
> >
> > 2. While I agree that “Markovian” is not an intrinsic property/requisite, the original MNO work goes much more beyond discussing Markovianity implied by the rebuttal.
> >
> > 3. The claim that the MNO paper is  'to learn the global invariant measure based on the current state' does not accurately reflect its contributions. As a key work of related research in 'long-term forecasting of dissipative chaos', the omission of a substantive discussion and comparison remains puzzling. Particularly given the 'spectral factor' claims, the scientific positioning of the manuscript is broadly related.
> >
> > I encourage the authors to incorporate these clarifications, polish and revise the manuscript rigorously.

---

> ### Author Response · Authors · 2025-11-22
> **Response to Reviewer ZSmM – Clarifying Positioning and Related Work (MNO, PFNN, Reservoirs)**
>
> The reviewer is right that proper attribution is important. While FERN targets chaos, stochastics dynamics and nonstationarity, and chaotic datasets serve primarily as diagnostic tools in our work rather than the main application domain, we should still properly contextualize relevant chaos forecasting literature, including MNO, PFNN, recent reservoir methods such as [Platt, 2023], zero-shot research [Zhang and Gilpin, 2025] and the diffusion work flagged by the reviewer. The discussion comparing FERN's geometry-preserving approach with MNO's physics-informed frameworks **is important**.
>
> &nbsp;
>
> We will ensure the revision reflects this. Main text will highlight key distinctions relevant to our contributions, with detailed discussion and nuances in the appendix. We will also write a succinct summary of changes as a follow-up thread to the reviewer.
>
> &nbsp;
>
> Our previous rebuttal focused narrowly on a concise and non-rigorous explanation of why *MSE-only* comparison between iterative one-step (MNO/PFNN) and direct multi-step (FERN/baselines) methods may be misleading and *why we didn't treat it as a primary baseline*. It is not meant to be a comprehensive review of the works, and would not do justice to nuances of dissipativity constraints, Sobolev and contraction losses. They should be given space in the literature review. We fully agree.
>
> &nbsp;
>
> Also, the reviewer is correct that the term 'Markov Operator' used previously risks conflating a math notion with the specific 'Markov Neural Operator (MNO)' architecture; **this is imprecise.** Our revision in the paper will adhere to rigorous standards the reviewer rightly advocates for.
>
> &nbsp;
>
> Reference:
> [1] Jason A Platt, Stephen G Penny, Timothy A Smith, Tse-Chun Chen, and Henry DI Abarbanel. Constraining chaos: Enforcing dynamical invariants in the training of reservoir computers. Chaos: An Interdisciplinary Journal of Nonlinear Science, 33(10), 2023.
>
> [2] Yuanzhao Zhang and William Gilpin. Zero-shot forecasting of chaotic systems. In The Thirteenth International Conference on Learning Representations, 2025.

---

### Official Review · Reviewer_BPhh · 2025-10-30

**Soundness:** 2
**Presentation:** 1
**Contribution:** 2
**Rating:** 2
**Confidence:** 3

**Summary:**

The motivation of this paper is twofold: (1) it introduces Forecasting with Ellipsoidal RepresentatioN (FERN) for long-term time-series forecasting and (2) contributes a new evaluation protocol. The proposed FERN model is geometry-aware that applies per-patch local linear transport using explicit spectral factors (eigenvectors/eigenvalues). This method transforms a Gaussian base distribution into a chain of local ellipsoids to predict the future geometry rather than the exact trajectory. The proposed evaluation metrics include Sliced Wasserstein Distance (SWD) and Effective Prediction Time (EPT) to address the limitations of pointwise metrics on chaotic systems. Experiments demonstrate that FERN outperforms baselines on several chaotic systems. The model also remains competitive on standard ETT and Weather benchmarks.

**Strengths:**

- The evaluation protocol is a substantive and timely contribution that correctly diagnoses blind spots in current LTSF practice—namely, overemphasis on noisy, quasi-periodic data and pointwise metrics. Accordingly, the authors propose concrete solution: geometry-aware (SWD) and stability (EPT) metrics with stress-testing on chaotic systems.
- The experiment evaluation is thorough and transparent. The authors provide comprehensive ablations and clear setup details in the appendix . A particularly valuable contribution is the identification of "recency bias" in standard validation splits, which addresses a common pitfall in training time-series models.

**Weaknesses:**

- The paper aggregates concepts from chaos theory, Koopman operator theory, optimal transport, and normalizing flows, but the authors fail to integrate these complex ideas into a unified framework and make the presentation hard to follow.
- The “Scope and Distinctions” section notes that FERN borrows NF/OT/Koopman language while not constituting any of these. The framework scope is ambiguous and unclear. Additional clarification should be included to support the spectral transparency claim.
- The technical description is frequently replaced with unhelpful analogies, which severely hinders the understanding of the paper.

**Questions:**

- What does the paper mean when it says it targets "conditional local geometry, not the dynamics"? But the proposed model is used for learning and forecasting the dynamics.
- There are many blank margins in the manuscript; please remove them to improve readability and length compliance.
- The paper’s organization needs attention. There is no conclusion, and the references are placed after the appendix.
- In Algorithm 1, the Ellipsoidal Transport layer uses a fixed, learnable $K$ matrix to *mimic complex value eigenvalues*. How is this $K$ matrix trained, and is it shared across all datasets or trained per dataset? What evidence supports the claim that this simple block structure is sufficient to capture "global components"  of diverse dynamical systems?

---

> ### Author Response · Authors · 2025-11-20
> **Response to Reviewer BPhh: Clarifying Framework Scope and "Conditional Local Geometry"**
>
> ## On "Writing and Math Frameworks"
> > The paper aggregates concepts ..., but the authors fail to integrate these complex ideas into a unified framework...
>
> >The “Scope and Distinctions” section notes that FERN borrows NF/OT/Koopman language while not constituting any of these. The framework scope is ambiguous and unclear. Additional clarification should be included to support the spectral transparency claim.
>
> >The technical description is frequently replaced with unhelpful analogies, which severely hinders the understanding of the paper.
>
> The reviewer is **correct**. On rereading with fresh eyes, we **acknowledge** the connections to *Koopman theory section* are distracting: They are **NOT** strict complex eigenvalues, and come at *non-trivial* cost of performance. The "rotate-stretch-rotate back" formula of $U \Lambda U^T$ is friendlier.
>
> &nbsp;
>
> The reviewer also asked:
> > In Algorithm 1, the Ellipsoidal Transport layer uses a fixed, learnable matrix to mimic complex value eigenvalues. How is this matrix trained, and is it shared across all datasets or trained per dataset? What evidence supports the claim that this simple block structure is sufficient to capture "global components" of diverse dynamical systems?
>
> The reviewer is sharp to question its training: these are per-dataset learnable parameters, and the complex block structure aimed to provide a data-specific oscillatory information, acting as a "general preconditioning". **YET**, we did not place a unitary constraint loss, we agree it adds complexity without commensurate performance gain.
>
> &nbsp;
>
> Our new plots (provided in a separate thread) demonstrates the $U \Lambda U^T$ and its spectral transparency is capturing the Lorenz63 well enough; we believe the fixed K layer can be discarded.
>
> &nbsp;
>
> **Action Taken**: We **remove the Koopman theory section from the paper**. New experiments were run on a model without Koopman components. We remove the "collaborative movie" / director analogy..
>
> &nbsp;
>
> **On ANF**: reviewer tQZD's description of **bidirectional coupling network that iteratively refines x and z ~ N(0,I)** is **exactly right**. We use ANF and Affine Coupling Layers (ACL) terminology as the most efficient way to describe this setup, but the nuances confuses readers.
>
> **Action Taken**: We **trim the ANF section** significantly. This leaves OT as the central story.
>
> ## On "local geometry"
> > What does the paper mean ... "conditional local geometry, not the dynamics"? But the proposed model is used for learning and forecasting the dynamics.
>
> **Traditional Dynamics**: Learn a vector field map F "rolling" *data* over time, **autoregressively**: we hope $F(y_{0:t}) \approx y_{t:2t}$ and $F(y_{t:2t}) \approx y_{2t:3t}$.
>
> &nbsp;
>
> **By Brenier's Theorem**, from abs.cont source such as t-dim N(0,I), **there exists a unique** OT from N(0,I) to true target y_{t:2t}, or any distribution (with finite second moments) produced by F. Similarly, when **we divided length t dim into patches** with patch size p, the OT's from p-dim N(0, I) to segments {0:p}, {p:2p} ... also **uniquely exist**.
>
> &nbsp;
>
> **Our Approach**: Learn **parallel** G "pushing" *N(0,I)* to true target, mimicking the unique SPSD solution by *approximating* SPSD Jacobian A. The Gaussian shape N(μ, A A^T) induced by A acting on N(μ,I) is the **conditional local geometry** of that patch.
>
> &nbsp;
>
> In short, **Dynamics vs Local geometry** is about using route F or route G:
>
> &nbsp;
>
> -F: formula is global and fixed across time, interpretable, hard to find, susceptible to nonstationarity. **Jacobian of F** is general: n-dim Jacobian has O(n^2) entries to search, expensive for **fully data-dependent** modeling.
>
> -G: data-dependent, geometrically interpretable, easier to find (SPSD is a much smaller function class), stable to nonstationarity, see new experiments in upcoming thread as proof.
>
> &nbsp;
>
> **The Preliminary Benefit**: *SPSD Jacobians* can be decomposed into $U \Lambda U^T$. **Fact: any n-dim rotational matrix U can be decomposed to *at most* n Householder products**. We can search low-rank, *fully* data-dependent Jacobian matrix with O(Rn) cost, where R=# of reflections. Eigenvalues and eigenvectors (for each patch) come "for free", without an O(n^3) decomposition cost. **NOTE:** the eigenvalues are not conflated by system eigenvalues: it is how *FERN* believes N(0,I) should stretch at that segment, so they are comparable across patches and rounds. See the new plots in an upcoming thread.
>
> &nbsp;
>
> **The Real Benefit**: prediction horizon=336, reflection R=8, a *low-rank* Jacobian matrix costs O(8\*336). Full rank costs: O(336^2). **With patching**, 336=14 size-24 patches, *full-rank* patch Jacobian cost O(24\*24), **trainable in parallel**. For dynamics F, we need to iteratively roll forward 14 times **sequentially**.
>
> &nbsp;
>
> *Action Taken*: Include rigorous definition. The motivations and benefit analysis directly included in the text.

---

> ### Author Response · Authors · 2025-11-21
> **Response2 to Reviewer BPhh: Non-Stationarity Experiments, Geometry Plots and Writing**
>
> We added **21 new experiments** on **non-stationarity** today, where FERN is shown to **not only dominate on chaotic datasets**, but also one of the **top performer in stochastic datasets** with different **controlled shocks**. Details are in posted new official comments. We invite the reviewer to review those new experiments.
>
> &nbsp;
>
> We added four **new plots** to demonstrate the **advantage** of the *explicit eigenvalue* approach: **when the data changes rapidly, the local max eigenvalues increase rapidly and the timing of the responses is very accurate.** It also shows where the model **fails**: on one side of attractor, the *overall eigenvalue responses (trace variation)* is more than necessary. Such diagnostics are available to users **despite an almost perfect reconstruction plot with very low MSE.**
>
> &nbsp;
>
> We thank the reviewer for noting:
> >The evaluation protocol is a substantive and timely contribution that correctly diagnoses blind spots in current LTSF practice—namely, overemphasis on noisy, quasi-periodic data and pointwise metrics.
>
> We believe the new experiments strengthen the evaluation protocol contribution: non-stationarity is no longer a relatively vague notion associated with "real-world" datasets such ETT; **the type, exact timing of non-stationarity are explicit, and the before and after distributions are transparent.** These experiments support the paper's central claim: the current major models **TimeMixer, PatchTST and DLinear,** despite performances on ETT, *perform poorly under explicit non-stationarity*. Interestingly, ModernTCN [ICLR 2024], a non-Transformer architecture, also shows strong performance, suggesting **architectural diversity matters for robustness**.
>
> &nbsp;
>
> We will provide a carefully revised PDF soon, with unhelpful analogies removed and include the full table of new experiments as well as the four new plots.

---

> > ### Comment · Reviewer_BPhh · 2025-11-24
> >
> > I would like to thank the authors for their detailed responses and extensive numerical validation. My point-by-point feedback is as follows:
> > - "On "Writing and Math Frameworks" I am pleased to see the planned revisions and look forward to the updated manuscript.
> > - "Local Geometry". The response clarifies the main idea behind the OT, in particular its link to Brenier’s theorem. I have one further question: it seems that different maps $G$ are fitted in parallel to different sequences/snapshots. How it can be used in practice? Can the method be used for extrapolation or forecasting, e.g., training up to $[nt,(n+1)t]$ and forecasting $[(n+1)t,(n+2)t]$? If relevant numerical experiments are already included, please point them out; if not, could the authors further comment on this aspect? Otherwise, the practical impact of the approach appears limited.
> > - Extras: It is great to see new evidence on the benefits of low-rank matrix structure and the performance on stochastic datasets.

---

> ### Author Response · Authors · 2025-11-25
> **Response 3 to reviewer BPhh: Local Geometry and Extrapolation**
>
> We sincerely thank the reviewer for the response. In the upcoming revision, to ensure a cleaner read, we have significantly streamlined the text by removing the Koopman/ANF analogies to focus purely on the Brenier motivation. We will send a succinct summary note to the reviewer on the changes made in response to the review. We are also happy to have the chance to engage in further discussion.
>
> &nbsp;
>
> The reviewer wrote
> > Can the method be used for extrapolation or forecasting, e.g., training up to $[nt, (n+1)t] and forecasting [(n+1)t, (n+2)t]? How it can be used in practice? Can the method be used for extrapolation or forecasting, e.g., training up to [nt, (n+1)t] and forecasting [(n+1)t, (n+2)t]?  If relevant numerical experiments are already included, please point them out; if not, could the authors further comment on this aspect? Otherwise, the practical impact of the approach appears limited.
>
> After carefully parsing the reviewer's question, we think we can pinpoint the issue.
>
> &nbsp;
>
> The term **local** refers to the conditional geometry given the current history window, not to time-specific fitting. For any input history window (e.g., x[2500:2800]), the encoder produces latent representation $z$, which then generates the patch-wise spectral factors ($U$, $\Lambda$, $t$) in the forecast for x[2800:3000]. These spectral factors are function outputs, **predicted conditionally** based on the input $x$. In other words, **the local Jacobians are network outputs (of mini-MLPs), not pre-fitted learnable parameters.**
>
> &nbsp;
>
> FERN can be used as standard autoregressive model for further prediction beyond the longest horizon (720 in our case). To predict future y[720:1440]: it takes current prediction y[0:720] as input, and asks the networks to conditionally generate parameters to forecast y[720:1440].
>
> &nbsp;
>
> For evaluation protocol, we strictly follow the standard LTSF setup where
> training, validation, and test sets are chronologically separated (e.g., Train:
> [0:1000], Val: [1000:1200], Test: [1200:1500]). The model never sees future
> data during training. The results in the paper, including the new experiments,
> demonstrate consistent extrapolation to chronologically separated test set,
> particularly the 100× improvement on Rössler against major forecasting models.
>
> &nbsp;
>
> We come back to the reviewer's first question.
>
> > it seems that different maps $G$, are fitted in parallel to different sequences/snapshots. How it can be used in practice?
>
> There is one **single shared network $G_\theta$** that produces instance-specific spectral factors. Given any input window, this network generates conditional parameters (U, Λ, t) for that specific input. The network weights are shared (learned globally), but the output parameters are instance-specific (generated per-window). We interpret the **deeper concern** as follows: since parameters are generated per-window rather than learned globally, how can we ensure robust generalization to unseen data?
>
> &nbsp;
>
> Per the ablation tests (Table 2) in the paper, *without the encoder*, forecast MSE on ETTH2 is 408.49--reduced to 11.19 *with encoder*. It confirms that the Gaussian latent $z$ is *by far* the most important reason driving the performances. We believe that the encoder's *bidirectional coupling network* (previously ANF, but we strip the term now) through training has already learned the underlying dynamics $F$ to a certain degree; its job is to encode the knowledge into a clean, compressed latent $z$, so the specific parallel form $U(z) \Lambda(z) U(z)^\top$ can work.

---

> > ### Comment · Reviewer_BPhh · 2025-11-27
> >
> > I thank the authors for their detailed clarifications.
> >
> > The explanation that “local” refers to conditional geometry given the current history window, and that the spectral factors are outputs of a single shared network, is helpful. It clarifies that FERN is used as an autoregressive model, with parameters generated per window to forecast the next segment.
> >
> > Given that parameters are generated per-window, it seems a short discussion on why this design still yields robust generalization to unseen data (e.g. tied to the role of the encoder and the latent representation) would also improve readability.
> >
> > I look forward to reading the updated manuscript and will finalize my evaluation once the new version is available.

---

> > > ### Author Response · Authors · 2025-11-27
> > > **Response 4 to Reviewer BPhh: Clarification on Autoregressive vs. Direct Forecasting**
> > >
> > > We appreciate the reviewer's response and would like to clarify one distinction:
> > >
> > > &nbsp;
> > >
> > > The reviewer wrote:
> > > > The explanation that “local” refers to conditional geometry given the current history window, and that the spectral factors are outputs of a single shared network, is helpful. It clarifies that FERN is used as an autoregressive model, with parameters generated per window to forecast the next segment.
> > >
> > > We would like to emphasize that *in the standard LTSF sense*, FERN is *not* an autoregressive model. The term is usually reserved for models such as RNN, or Transformer variants, where **within the window**, say y[0:200], later elements depend on prediction from earlier elements, like y[196] is generated by feeding the model the previous prediction y[195], which is slow and inefficient. FERN is *multi-step direct model*, where it *generate the entire window* [y0:200] in one go. It divides the window into patches, but predictions for all patches are generated in parallel.
> > >
> > > &nbsp;
> > >
> > > Our previous clarification meant to say, if the user would like to see the **out of window future predictions solely conditional on the current input x**, i.e. without feeding the true y[0:200] values to predict y[200:400] but rely on current x (the immediate data before y[0:200]), we *can* make it an autoregressive model.
> > >
> > > &nbsp;
> > >
> > > The reviewer also writes:
> > > > Given that parameters are generated per-window, it seems a short discussion on why this design still yields robust generalization to unseen data (e.g. tied to the role of the encoder and the latent representation) would also improve readability.
> > >
> > > We thank the reviewer for the suggestion and we fully agree. We will add dedicated short discussion on generalization specifically. We appreciate the reviewer's constructive engagement and welcome any further questions and suggestions as we revise the manuscript.

---

### Official Review · Reviewer_hekH · 2025-10-31

**Soundness:** 3
**Presentation:** 2
**Contribution:** 2
**Rating:** 4
**Confidence:** 4

**Summary:**

The paper proposes FERN, a geometry-aware forecaster that represents future time-series patches as locally linear ellipsoids. It introduces a new evaluation protocol using Wasserstein distance and Effective Prediction Time, arguing that existing long-term forecasting (LTSF) metrics overfit noise and miss chaos.

**Strengths:**

I like the theoretical explaination of this article, looks good

**Weaknesses:**

- Writing is heavy, overly theoretical, and sometimes reads like a position essay rather than a reproducible model paper.
- No phase-space or attractor plots to prove that geometry preservation actually happens. Figure 1 is schematic only.

**Questions:**

1. Add visual reconstructions of chaotic attractors showing ellipsoidal chains.
2. How does the model sensity to the initial condition? some demonstration would be greatful, it is not clear to me.

---

> ### Author Response · Authors · 2025-11-18
> **Response to Reviewer hekH: Writing Clarity and Interpretability Visualization**
>
> We sincerely thank the reviewer for their effort and thoughtful comments.
>
> ## On Writing
> > Writing is heavy, overly theoretical, and sometimes reads like a position essay rather than a reproducible model paper.
>
> **Yes**, we **acknowledge** the writing needs improvement. We think **Koopman theory section** should belong to another paper. The fixed, learnable parameters included are not strict complex eigenvalues, providing limited additional interpretability at *non-trivial* cost of performance. The easy "rotate-stretch-rotate back" formula of $U \Lambda U^T$ is much easier to read.
>
> **Action Taken**: We **remove the Koopman theory section from the paper** and moved the connection discussion to appendix. The new experiment data is tested on a model without Koopman components.
>
> **Action Taken**: We include a set of new experiments (upcoming in a separate thread), which to our knowledge, is the first set of systematically study on controlled nonstationarity. *Positional framing* will be reduced, replaced with empirical results and plots to **demonstrate** our point.
>
> ## On Plot
> >No phase-space or attractor plots to prove that geometry preservation actually happens. Figure 1 is schematic only.
>
> &nbsp;
>
> **Action Taken**: We include three plots in our anonymized GitHub repository. We recommend opening a new browser tab. For a single place to access, we prepare anonymous uploads at imgur as well.
>
> https://imgur.com/a/UcXsjPj
>
> &nbsp;
>
> - **Plot 1:** Side-by-side Lorenz–63 reconstruction with data vs FERN prediction.
> https://anonymous.4open.science/api/repo/FERN-1F63/file/attr_plot.png?v=eeed31d6
> &nbsp;
> https://imgur.com/a/3E8X8dB
>
> - **Plot 2:** Single attractor reconstruction with color-coded local behavior.
> https://anonymous.4open.science/api/repo/FERN-1F63/file/attr_plot_2.png?v=04046b74
>
> - **Plot 3:** Error analysis and an eigenvalue “heatmap” over time.
> https://anonymous.4open.science/api/repo/FERN-1F63/file/diag_plot.png?v=162c00e5
>
> - **Plot 4:** Similar to Plot1, but display **largest** eigenvalues, **not** sum of eigenvalues
> https://anonymous.4open.science/api/repo/FERN-1F63/file/diag_plot2.png?v=34a505c5
>
> &nbsp;
>
> **Setup**: This is a reconstruction of a prediction of Lorenz63, 25k samples, seed=7, input sequence length=336 and prediction horizon=96. The MSE is about 1.1 (twice as large as the reported average MSE in the table, due to model changes). **YET**, with FERN’s design, we can describe exactly **where Fern succeeds and fails, despite the low MSE loss**:
>
> &nbsp;
>
> **Analysis**: In plot 1, the **left panel** shows *local speed* (magnitude of first differences). **On the right**, it describes the *normalized trace* (sum of eigenvalues Σλᵢ of transformation matrix A), representing total uncertainty. Higher values (darker colors) indicate the
> model is 'guarding heavily' by widening its ellipsoids.
>
> &nbsp;
>
> Clearly, *on this trajectory*, the system has highest speed on the outer lobes of left and right wings; On the inner, especially lower half of each lobe, the speed is *very slow*, leaving the gradient map almost white.
>
> &nbsp;
>
> **Where Fern succeeds**: the general structure is accurate. On the *slow part*, the eigenvalues are also very low, leaving the gradient map almost white. Clearly, *the slow part and general structure is well captured*.
>
> &nbsp;
>
> **Where Fern fails**: the gradient map is *considerably darker on the right lobe*. The model is *over-expanding uncertainty on the right lobe*, inflating uncertainty on the right lobe. On the left wing, it is far more confident.
>
> &nbsp;
>
> In plot 3, we flatten the first state variable (x in Lorenz–63) and examine errors together with eigenvalue activity. **NOTE:** we display **max eigenvalues (spectral radius)**, not sum of eigenvalues. The log determinant (volume changes) are also available, but it and trace are not as indicative.
>
> &nbsp;
>
> We move onto plot 4, the same side by side comparison now displays **max eigenvalues**. The gradient map is *significantly closer*, the right wing sensitivity disappears. The insight: problem is not with max eigenvalues, but smaller eigenvalues activity on the right wing.
>
> &nbsp;
>
> We observe that:
> - The largest eigenvalue (spectral radius) activations cluster around **violent “top–bottom–top” swings** of the trajectory.
> - The worst prediction errors also align with these extreme flips (with some timing offset).
>
> **Conclusion**: Taken together, these plots show that **even at low MSE ≈ 1/8 of the empirical standard deviation**, FERN’s ellipsoids provide *actionable interpretability*: they localize where the model is confident and where it is “overcautious” or unstable. Spectral radius alone tells what the model think is important. The eigenvalues are soft-clipped to 4.5, but users can tune for each dataset based on feedback like this.

---

> > ### Comment · Reviewer_hekH · 2025-11-21
> >
> > Thanks for your reply, I've adjusted my score to 6

---

> > > ### Author Response · Authors · 2025-11-21
> > >
> > > We thank the reviewer for taking the time to review the comment. We are also deeply grateful to the score adjustment. We added 21 new experiments on non-stationarity today, where FERN is shown to not only dominating in chaotic dataset, but also the one of the top performer in stochastic datasets with different controlled shocks. Details in posted new official comments. We invite the reviewer to review those new experiments.

---

### Official Review · Reviewer_tQZD · 2025-11-01

**Soundness:** 2
**Presentation:** 2
**Contribution:** 2
**Rating:** 2
**Confidence:** 4

**Summary:**

The authors propose FERN, a model for long-term time series forecasting that represents local dynamics via ellipsoidal approximations with symmetric positive semi-definite Jacobians. The paper proposes a new evaluation protocol stress-testing on chaotic systems using Wasserstein-2 distance and effective prediction time metrics. The methodology aims to predict local conditional geometry via first-order Taylor approximation coupled with spectral decomposition, along with a Koopman-inspired global operator layer. On chaotic benchmarks, FERN achieves significantly better performance while remaining competitive on standard ETT benchmarks.

**Strengths:**

The paper achieves strong empirical results on chaotic benchmarks, with FERN substantially outperforming baselines. This performance gain, regardless of the theoretical framing issues, demonstrates that the specific architectural choices have merit for certain types of time series.

The emphasis on stress-testing models on chaotic systems is valuable, even if not entirely novel. The paper makes a reasonable argument for why current benchmarks may reward overfitting to specific historical trajectories rather than learning generalizable dynamics.

The Takens' embedding discussion, while not directly justifying the proposed method, provides useful context for understanding why simple models succeed on some benchmarks and why patching works well.

**Weaknesses:**

- The core idea of the FERN model is conceptually interesting but it is presented as an ad-hoc combination of normalizing flow, optimal transport, and Koopman frameworks. The authors explicitly state they are "not actually" implementing any of these theories rigorously, suggesting design-by-analogy rather than derivation from first principles and it also raises the question of why these frameworks are invoked at all.

- I found the collaborative movie adaptation analogy in Section 3 unhelpful and it made the technical content hard to parse. My understanding is that the proposed FERN model consists of two components:
   - **Encoder:** Despite being framed as "adapted ANF," this is simply a bidirectional coupling network that iteratively refines x and z ~ N(0,I) as follows: For i = 1…5: $z \leftarrow s^*(x) \odot z + t(x)$, $x \leftarrow s^*(z) \odot x + t(z)$. This mapping lacks the fundamental properties of normalizing flows and I don't see any clear benefit from the ANF framing.
  - **Ellipsoidal Transport:** The prediction is computed as: $y^* = U(z) K \Lambda (z) K^T U(z)^T y_0 + t(z)$, where $y_0 ~ N(0,I)$. This is a structured linear transformation of Gaussian noise, conditioned on z. The optimal transport framing does not provide any meaningful insight since there is no transport problem being solved, no source/target distributions, and no transport cost. The Brenier theorem citation is irrelevant since this is supervised learning with MSE loss, not measure transportation. The SPSD structure merely constrains the linear map's form but doesn't make it an OT map.

  In summary, FERN essentially learns a nonlinear mapping from input x to parameters of an affine transformation applied to Gaussian noise, trained with standard MSE loss. The extensive discussion of ellipsoids, optimal transport, and Koopman theory obscures rather than illuminates this simple architecture. The actual contribution appears to be the specific parameterization choices that is shown to work well empirically on chaotic systems, not any deep connection to the invoked mathematical frameworks.

- I was unable to find a formal definition of local conditional geometry and it was not clear to me if the ellipsoids refer to conditional distributions or prediction sets or second-moment approximations.

- The paper does not include a conclusions section and ends abruptly with the numerical studies.

- The results for chaotic systems are impressive. However, the discussion of results needs more depth. For instance, it was not clear what happens beyond a few Lyapunov times where pointwise prediction is theoretically impossible.

**Questions:**

- Please formally define what "local conditional geometry” means in the context of this work.
- Section 3.2 provides a W2 bound; however, here deterministic predictions are being made and not distribution matching. Can you clarify what distributions are being transported and how this relates to minimizing MSE on point predictions?
- Algorithm 1 shows U(z) is data-dependent. How do you parametrize U to maintain orthogonality during SGD?
- How do you prevent rank collapse?
- How does eigendecomposition cost scale with patch size P, number of patches L, and horizon H? It will be useful to provide wall-clock time comparisons at varying scales (e.g., d = 1, 10, 50; T = 100, 1000, 10000).
- Can you please provide details of the data preprocessing protocol used for FERN in the numerical studies.

---

> ### Author Response · Authors · 2025-11-13
> **Response: Clarification of the Brenier OT Map as a Design Principle**
>
> We sincerely thank the reviewer for his thorough reading of the paper and enthusiastic and engaging comments.
>
> While we are working to solidify the writing, we think a few **preliminary response** over **mathematical questions** would be beneficial for the discussion.
>
> Revisions, further comments and more experiments are underway.
>
> &nbsp;
>
> # On "Transport andOT map" (emphasis ours)
> > The optimal transport framing does not provide any meaningful insight since there is **no transport problem being solved**, **no source/target distributions**, and **no transport cost** ... The SPSD structure merely constrains the linear map's form but **doesn't make it an OT map**.
>
> &nbsp;
>
> **We respectfully, but firmly, disagree on the mathematical front**.
>
> &nbsp;
>
> **The Source**: explicitly, the Gaussian noise $y_0 \sim N(0, I)$
>
> &nbsp;
>
> **The pushforward**: Let $T(y_0) = μ + A y_0$, then $T_* N(0, I)=N(μ, A A^T)$ **is a transport map**.
> Here A is *symmetric and positive semi-definite by design*, as $A = U(z) diag(σ) U(z)^\top$ and constant w.r.t. y_0.
>
> &nbsp;
>
> **Target distribution**: $N(μ, A A^\top)$.
>
> &nbsp;
>
> **Cost**: quadratic cost $\( c(y_0, y) = \|y_0 - y\|^2 \)$.
>
>
> &nbsp;
>
> Crucially, for Gaussians, $μ + A y_0$  is exactly the OT, i.e. **unique gradient map with SPSD Jacobian from $N(0, I)$ to $N(μ, A A^\top)$**, because A is *the* symmetric square root  of $A A^\top$, not arbitrary SPSD matrix. This holds regardless of whether you train with NLL, MSE or SWD.
>
> &nbsp;
>
> The general case also applies: the SPSD constraint always **induce** an OT from y_0 to the *induced, predictive* distribution $Ay_0$ (i.e. $T_* N(0, I)$), due to the uniqueness of OT.
>
> &nbsp;
>
> # On "local conditional geometry"
>
> &nbsp;
>
> **Local geometry** = the Gaussian shape obtained by pushing forward y_0~$N(0,I)$ to $N(μ, A A^\top)$.
>
> &nbsp;
>
> # On "Connection to Invoked Mathematical Framework" (emphasis ours)
> > ... it is presented as an **ad-hoc combination**, ...  ,**suggesting design-by-analogy rather than derivation from first principles**..., and it also **raises the question of why these frameworks are invoked at all**, ... , **not any deep connection to the invoked mathematical frameworks**.
>
> **We present a short argument on why Brenier's is central to our model**.
>
> **Traditional Dynamics**: Learn a vector field map F "pushing" *observed states* over time: given $y_{0:t}$, we hope $F(y_{0:t} \approx y_{t:2t}$. By Brenier's Theorem, under abs.cont source **there exists unique** OT's transporting t-dim $N(0,I)$ to either true target $y_{t:2t}$, or the dynamical best prediction $F(y_{0:t})$ (even for delta function, OT handles them). Similarly, when we divided length t dim into patches with patch size p, the OT from p-dim $N(0, I)$ to segments $y_{0:p}, y_{p:2p}$ ... also **uniquely exists**.
>
> &nbsp;
>
> **Our Approach**: Learn G "pushing" *noise* to true target: for each patch, we hope $G_p(N(0, I)) \approx y_{t:t+p}$. For each patch, we have SPSD Jacobian A, and the Gaussian shape $N(μ, A A^\top)$ induced by A acting on N(0,I) is the **local geometry** of that patch.
>
> &nbsp;
>
> **Dynamics vs Local geometry**:
>
> -F: formula is global and fixed across time, accurate, interpretable, hard to find, susceptable to nonstationarity. **Jacobian** is general: n-dim Jacobian has O(n^2) entries to search, for **fully data-dependent** modeling.
>
> -G: data-dependent, approximation, interpretable as Gaussian Ellipsoids, comparable eigenvalues, easier to find, stable to nonstationarity (new experiments as proof). **SPSD Jacobians** can be decomposed into U diag() U^T, where U is orthogonal and constructed as R Householder reflections (each O(n) cost). *Fact: any n-dim rotational matrix can be decomposed to **at most** n Householder products;* Therefore, **without patching**, we can search *low-rank, fully data-dependent* Jacobian matrix with O(Rn) cost. Impossible with general F().
>
> &nbsp;
>
> **The Real Benefit**: **Without patching**, for a 336-dim prediction horizon, if we set number of reflection R=8, we can find a *low-rank* Jacobian matrix at O(8\*336) cost. The full rank search is still O(336^2). **With patching**, let p = 24 so we have 14 patches, the we can search the *full-rank* patch Jacobian at O(24\*24) cost, **in parallel**.
>
> &nbsp;
>
> On the other hand, we *target* the dynamics, then we can only find the *full-rank*, no **data-dependent** low rank version is allowed, and we must do it **in sequence**, iteratively rolling forward 14 times.
>
> &nbsp;
>
> -Summary: Our trick depends on seemingly arbitrary SPSD constraint and only has gradient-like behaviour. A smooth F can be decomposed into gradient-like + rotational parts. **Brenier theorem critically guarantees we can do without rotational parts. The **empirical edge** on Lorenz over PatchTST (Table 1: 0.04 vs 8.33 MSE on Rössler) shows this on systems with **known complex rotational dynamics**. Parsimony outperforms unconstrained complexity.
>
> (Nov 18: rewrite for better readability.)

---

> ### Author Response · Authors · 2025-11-20
> **Reply 2: On the Relevance of Optimal Transport to MSE Training**
>
> We continue discussing the concern raised by the reviewer.
>
> ## On MSE and W2 (emphasis ours)
> >The Brenier theorem citation is **irrelevant since this is supervised learning with MSE loss, not measure transportation**. The SPSD structure merely constrains the linear map's form but doesn't make it an OT map...
>
> &nbsp;
>
> >Section 3.2 provides a W2 bound; however, **here deterministic predictions are being made and not distribution matching**. Can you clarify what distributions are being transported and how this relates to minimizing MSE on point predictions?
>
> After carefully parsing the review many times, we believe we can pinpoint the issue. First, we **acknowledge** the reviewer's **valid critique** regarding the W2 bound discussion: we use general language rather than concrete setting (empirical measure for Y, Gaussian source $y_0$, MSE-only training).
>
> &nbsp;
>
> Between prediction and target δ_y:  $W_2^2(N(\mu, \Sigma), \delta_y) = \|\mu - y\|_2^2 + \operatorname{tr}(\Sigma)$. MSE is the first term. We mean: SPSD matrix A *is* the OT pushing N(μ,I) to N(μ,AA^T=Σ(x)). **While we do NOT minimize tr Σ(x), with A, we did not increase tr Σ(x) unnecessarily either**, and tr Σ(x) is the variance structure needed by the model to get lowest MSE. This is geometry-preservation argument presented poorly.
>
> &nbsp;
>
> **Action Taken**: We delete this passage.
>
> &nbsp;
>
> **Yet**, we think the **deeper question** *actually* being asked is, **why invoke OT when using only MSE?** our motivation is:
>
> &nbsp;
>
> 1. We want to have a *unified framework* for chaos, stochastics, and nonstationarities.
>
> &nbsp;
>
> 2. We are doing *predictions* and we care about pointwise metrics e.g. MSE. The true distribution is *not only unknown but changing*, so we focus on *empirical measure*, also the right level for chaotic datasets. *This is a prediction paper, NOT OT paper.*
>
> &nbsp;
>
> 3. In e.g. MSE only training, we approach solutions where MSE is minimized while
> tr Σ(x) varies. Per Brenier, one *unique, MSE=0, minimal tr Σ(x) solution* **always exists** in that solution set, **with SPSD Jacobian.**
>
> &nbsp;
>
> 4. So, let F be **arbitrary dynamics** that "rolls" data along time axis, and G be the **SPSD map** transporting from a N(0,I). **With MSE only training + SPSD Jacobian matrix search**, any distribution (with finite second moments) produced by F is exactly matched by nonlinear true OT.  FERN (G) approximates this transport by matching the target with an optimal Gaussian (Ellipsoid), as our map is piece-wise linear.
>
> &nbsp;
>
> The **key difference** is not in what distributions can be reached, but in *how*: from the **sequential, autoregressive** evolution of F to a **parallel movement from a unified Gaussian source** in the transport-map style of G.
>
> &nbsp;
>
> So, this is a "no expressiveness loss with small function class" argument,  and OT was invoked so Brenier's theorem handles both dynamics. We gain computational efficiency
> (O(Rn) vs O(n²)) and interpretability (eigendecomposition UΛU^T).
>
> **Action Taken:** We include the motivation *directly* in the intro.
>
>
> ## On "ANF framing and Koopman theory"
>
> > The extensive discussion of ellipsoids, optimal transport, and Koopman theory **obscures rather than illuminates** this simple architecture.
>
> >...This mapping lacks the fundamental properties of normalizing flows and I don't see any clear benefit from the ANF framing.
>
> The reviewer's description of **bidirectional coupling network that iteratively refines x and z ~ N(0,I)** is **exactly right**. We only use ANF and Affine Coupling Layers (ACL) terminology since they are the most efficient description.
>
> &nbsp;
>
> We **also agree with the reviewer** that *Koopman theory section* belongs to another paper. The fixed, learnable parameters included are **NOT** strict complex eigenvalues, and come at *non-trivial* cost of performance. The "rotate-stretch-rotate back" formula of $U \Lambda U^T$ is friendlier.
>
> **Action Taken**: We **remove the Koopman theory section from the paper**. New experiments were run on a model without Koopman components. We **trim the ANF section** significantly, details moved to appendix. This leaves OT as the central story.
>
>
> ## On "orthogonality and rank collapse"
> We ask NN to generate unit reflection vectors v and apply the Householder reflections to data x via [
> Hx = I - 2 v v^\top x
> ] An even number of Householder reflections produces a pure rotation, each with O(n) cost.
>
> &nbsp;
>
> So, orthogonality is guaranteed *by construction* and unaffected by training regime. We do not need e.g. the costly O(n^3) Cayley parametrization. Rank collapse is controlled by the eigenvalues Λ, not the orthogonal matrix U. Here are some ablation tests.
>
> &nbsp;
>
> Reflection: 8
> MSE=1.109 | MAE=0.570
> time: 484.86 seconds
>
> &nbsp;
>
> Reflection: 24 train with 3 size-8 blocks (updates one third each forward)
> MSE=1.286 | MAE=0.628
> time: 457.08 seconds
>
> &nbsp;
>
> Reflection: 24
> MSE=1.707 | MAE=0.662
> 471.50 seconds

---

> ### Author Response · Authors · 2025-11-22
> **Response 3 to Reviewer tQZD: New Experiments, Lyapunov Horizon, Scalability, Data Processing**
>
> We added four **new plots** to show the **advantage** of having *explicit eigenvalues*: **when the data changes rapidly, the local max eigenvalues increase rapidly and the timing of the responses **tracks** changes accurately.** It also shows where the model **fails**: on one side of the attractor, the *overall eigenvalue response (trace variation)* is more than necessary. Such diagnostics are available to users **despite an almost perfect reconstruction plot with very low MSE.** Details are in the posted official comments.
>
> &nbsp;
>
> We posted results of **21 new experiments** on **non-stationarity** today, where FERN is shown to **not only dominate on chaotic datasets**, but also one of the **top performers in stochastic datasets** with different **controlled shocks**. Details are in the newly posted  official comments. We invite the reviewer to look at those new experiments.
>
> The reviewer wrote:
> > This performance gain ... demonstrates that the specific architectural choices have merit for certain types of time series.
>
> In terms of MSE and SWD ranks (with equal weight), FERN is the **best on both chaotic and stochastic datasets**. For example, even without injected non-stationarity, it has the best MSE on **Seasonal AR, GARCH, Ornstein–Uhlenbeck** processes, indicating FERN's effectiveness extends beyond chaotic systems to general-purpose forecasting.
>
> &nbsp;
>
> ## On Lyapunov time
> > For instance, it was not clear what happens beyond a few Lyapunov times where pointwise prediction is theoretically impossible.
>
> Lorenz 63's Lyapunov time is about 1.1, so 720 steps with dt=0.01 is about 6.5 Lyapunov time -- initial errors have grown by a factor of e^6.5. As shown in the Lorenz63 table in Appendix, FERN's advantages in MSE are erased, converging to to other models. **At this point**, however, MSE are likely not indicative of model capability.
>
> &nbsp;
>
> Yet, as claimed in the paper, FERN has the best shape fidelity: SWD is 4.89 while TimeMixer and PatchTST in 10's and DLinear is at 40.
>
> &nbsp;
>
> **Action Taken:** We include this in the upcoming PDF.
>
> ## On "Eigendecomposition Scalability":
> >How does eigendecomposition cost scale with patch size P, number of patches L, and horizon H? It will be useful to provide wall-clock time comparisons at varying scales (e.g., d = 1, 10, 50; T = 100, 1000, 10000).
>
> We provide wall-clock time for H=96, 720 and 1200. We note that horizons beyond H=1000 are uncommon in LTSF benchmarks and chaotic systems become unpredictable beyond ~10 Lyapunov times.
>
> **Theoretical complexity:** Let horizon H=L\*P, then as mentioned, since we do not implement eigendecompositions with O(H^3) costs,
> for *no patch, low-rank representation* we have O(R\*H) cost; for L patches it is O(R\*L\*P)=O(R\*H).
>
> &nbsp;
>
> This is because we **construct** O(H) eigenvalues and O(R\*H) eigenvectors *to get our SPSD matrix.* For feature dimension d, our channel-independent architecture scales linearly by reshaping to [B×D, 1, H].
>
> &nbsp;
>
> **Empirical timing (ETTH2, patch_size=24, input_len=336):**
> - H=96: 226.66s training time
> - H=720: 154.86s training time
> - H=1200: 190.63s training time
>
> The roughly constant training time (despite 12.5× horizon increase) confirms no pathological scaling.
>
> &nbsp;
>
> From our experience, the hidden layers and the number of stacks are usually the biggest bottleneck. After submission, we have done some additional searches, and *have cut the number of layers in hidden network by half.*
>
> ### On "Data Preprocessing"
>
> In the paper, we did not do any preprocessing.
>
> &nbsp;
>
> The reviewer however is **right to be concerned about this**. Since the paper's submission, our view evolved, as we identified *data quality issues* in ETT datasets. There exist several sentinel values (e.g. `88.297`) that stay constant for over a month, likely a failed sensor. Moreover, if we look at the zero distribution:
>
> Column | Total Zeros | % Zeros | len=1 Zero| len>1 Zeros|
> ------ | ----------- | ------- | --------- | -----------|
> HUFL   | 58          | 0.33%   | 1         | 57         |
> HULL   | 3836        | 22.02%  | 163       | 3673       |
> MUFL   | 0           | 0.00%   | 0         | 0          |
> MULL   | 1067        | 6.13%   | 245       | 822        |
> LULL   | 5792        | 33.25%  | 414       | 5378       |
> OT     | 1           | 0.01%   | 1         | 0          |
>
> Several columns are heavily zero-inflated. Standard pointwise losses such as MSE, MAE (i.e. Gaussian / Laplace error assumption) are **NOT** equipped for mixture of continuous and “structural” zeros, i.e. *zero-inflated distribution*. Also we observe long *consecutive* zero runs, which are more plausibly missing-data artifacts than true physical zeros.
>
> We propose a principled preprocessing scheme applied uniformly to all methods  (ours and baselines), with details in the appendix of upcoming PDF. This addresses the reviewer's question while maintaining fair comparison.

---

### Author Response · Authors · 2025-11-21
**Additional Results: 21 Controlled Non-Stationary Synthetic Benchmarks**

# NEW EXPERIMENTS

We ran 21 new experiments (336-in-336-out, 2 seeds, 25k-35k samples, 70%:20%:10% splits) to systematically investigate the effect of non-stationarity, **on both chaotic AND stochastic datasets**, and **we invite the reviewer to review the executive summaries.** Full table available in upcoming revised PDF.

&nbsp;

**Purpose of Additional Experiments**:

&nbsp;

- **Prove** FERN is not a chaotic specialist.

&nbsp;

- *Systematically* use the **synthetic datasets** to see how models handles (1) **chaotic** dynamics (2) **stochastic** dynamics and (3) various non-stationarities, where the **exact non-stationarities are known, explicit, precise.**

&nbsp;

We design three non-stationarities that **happen at *exactly* the midpoint of the train data:**

&nbsp;

-**Parameter drift**: system parameters change mid-way: e.g. Lorenz **parameters are [sigma=10, rho=28, beta=8/3]** and having generated half of the train data, **parameters become [sigma=10.1, rho=28.1, beta=8.1/3]** and keep generating the rest.

&nbsp;

-**State perturbation**: e.g. in Lorenz, **0.9 is added to every dimension**. This answers **reviewer hekH's question: what is model's sensitivity to initial conditions**. Due to states being perturbed, the Lorenz system is changed, and all models compete fairly under this setting.

&nbsp;

-**Regime replacement**: at midway, the trajectory is replaced by one generated with **different parameters and initial conditions**.

&nbsp;

We include three more models: **Koopa** [NeurIPS 2023], **ModernTCN** [ICLR 2024], **PFNN (Poincaré Flow)** [ICLR 2025] (**requested by reviewer ZSmM**).

&nbsp;

Summaries:

&nbsp;

**Average rank** (of MSE and SWD) across all tasks (lower is better):
-Fern: ≈ 1.95
-ModernTCN: ≈ 3.43
-TimeMixer: ≈ 3.57
-PatchTST: ≈ 3.93
-DLinear: ≈ 4.05
-Koopa: ≈ 4.55
-PFNN: ≈ 6.38

&nbsp;

-**FERN is best/co-best** in average MSE and SWD ranks **on 13/21 tasks**: **7/10 for chaotic, 6/11 on stochastic ones**. Other than **100x** better performance on Rossler, we see for example **48% reduction** of MSE against TimeMixer **on the stochastic classic**: Switching linear dynamical systems.

&nbsp;

Using TimeMixer as baseline, **Fern is**:

- **Switching LDS – base (no shock):** 1st in SWD, **13% lower SWD**, similar MSE.

- **Switching LDS – parameter change:** 2nd in SWD, **9% lower MSE, 14% lower SWD**.

- **Switching LDS – regime replacement:** 1st in both, **48% lower MSE, 58% lower SWD**.

- **Seasonal AR – base:** 1st in MSE, 2nd in SWD, **same MSE, 15% lower SWD**.

- **Seasonal AR – parameter drift:** 1st in MSE, 2nd in SWD, **2% lower MSE, 18% lower SWD**.

- **GARCH(1,1) – parameter shock:** 1st in MSE, **5% lower MSE**, but **12% higher SWD**.

- **Double-well SDE – base:** 1st in SWD, **8% lower MSE, 35% lower SWD**.

- **Double-well SDE – parameter shock:** 1st in both, **37% lower MSE, 40% lower SWD**.

- **Ornstein–Uhlenbeck SDE – base/param:** 2nd in MSE with **5–7% lower MSE**, but worse SWD; these Gaussian cases are easy and favor linear models.

- **Rössler – base:** 1st in both, **98% lower MSE, 99% lower SWD**.
- **Rössler – parameter shock:** 1st in both, **≈99% lower MSE and SWD**.

- **Lorenz-63 – base:** 1st in both, **50% lower MSE, 56% lower SWD**.
- **Lorenz-63 – state shock:** 1st in both, **≈61% lower MSE, 64% lower SWD**.
- **Lorenz-63 – parameter shock:** 1st in both, **≈52% lower MSE, 57% lower SWD**.

- **Lorenz-96 – base:** 1st in both, **35% lower MSE, 55% lower SWD**.
- **Lorenz-96 – regime replacement:** 1st in both, **20% lower MSE, 41% lower SWD**.

- **Chua’s circuit – base:** ModernTCN is 1st; Fern is 2nd with **40% lower MSE, 28% lower SWD** vs TimeMixer.
- **Chua’s circuit – parameter shock:** TimeMixer is 1st; Fern is 2nd with **≈14% lower SWD** and **≈2% higher MSE** vs TimeMixer.

---

### Author Response · Authors · 2025-11-21
**NEW PLOTS: Lorenz-63 Geometry and Eigenvalue Diagnostics markdown Copy code**

# NEW PLOTS

**Action Taken**: We include three plots in our anonymized GitHub repository. We recommend opening a new browser tab. For a single place to access, we prepare anonymous uploads at imgur as well.

https://imgur.com/a/UcXsjPj

&nbsp;

- **Plot 1:** Side-by-side Lorenz–63 reconstruction with data vs FERN prediction.
https://anonymous.4open.science/api/repo/FERN-1F63/file/attr_plot.png?v=eeed31d6
&nbsp;
https://imgur.com/a/3E8X8dB

- **Plot 2:** Single attractor reconstruction with color-coded local behavior.
https://anonymous.4open.science/api/repo/FERN-1F63/file/attr_plot_2.png?v=04046b74

- **Plot 3:** Error analysis and an eigenvalue “heatmap” over time.
https://anonymous.4open.science/api/repo/FERN-1F63/file/diag_plot.png?v=162c00e5

- **Plot 4:** Similar to Plot1, but display **largest** eigenvalues, **not** sum of eigenvalues
https://anonymous.4open.science/api/repo/FERN-1F63/file/diag_plot2.png?v=34a505c5

&nbsp;

**Setup**: This is a reconstruction of a prediction of Lorenz63, 25k samples, seed=7, input sequence length=336 and prediction horizon=96. The MSE is about 1.1 (twice as large as the reported average MSE in the table, due to model changes). **YET**, with FERN’s design, we can describe exactly **where Fern succeeds and fails, despite the low MSE loss**:

&nbsp;

**Analysis**: In plot 1, the **left panel** shows *local speed* (magnitude of first differences). **On the right**, it describes the *normalized trace* (sum of eigenvalues Σλᵢ of transformation matrix A), representing total uncertainty. Higher values (darker colors) indicate the
model is 'guarding heavily' by widening its ellipsoids.

&nbsp;

Clearly, *on this trajectory*, the system has highest speed on the outer lobes of left and right wings; On the inner, especially lower half of each lobe, the speed is *very slow*, leaving the gradient map almost white.

&nbsp;

**Where Fern succeeds**: the general structure is accurate. On the *slow part*, the eigenvalues are also very low, leaving the gradient map almost white. Clearly, *the slow part and general structure is well captured*.

&nbsp;

**Where Fern fails**: the gradient map is *considerably darker on the right lobe*. The model is *over-expanding uncertainty on the right lobe*, inflating uncertainty on the right lobe. On the left wing, it is far more confident.

&nbsp;

In plot 3, we flatten the first state variable (x in Lorenz–63) and examine errors together with eigenvalue activity. **NOTE:** we display **max eigenvalues (spectral radius)**, not sum of eigenvalues. The log determinant (volume changes) are also available, but it and trace are not as indicative.

&nbsp;

We move onto plot 4, the same side by side comparison now displays **max eigenvalues**. The gradient map is *significantly closer*, the right wing sensitivity disappears. The insight: problem is not with max eigenvalues, but smaller eigenvalues activity on the right wing.

&nbsp;

We observe that:
- The largest eigenvalue (spectral radius) activations cluster around **violent “top–bottom–top” swings** of the trajectory.
- The worst prediction errors also align with these extreme flips (with some timing offset).

**Conclusion**: Taken together, these plots show that **even at low MSE ≈ 1/8 of the empirical standard deviation**, FERN’s ellipsoids provide *actionable interpretability*: they localize where the model is confident and where it is “overcautious” or unstable. Spectral radius alone tells what the model think is important. The eigenvalues are soft-clipped to 4.5, but users can tune for each dataset based on feedback like this.

---

### Author Response · Authors · 2025-12-02
**Final update on ablation experiments.**

We have **uploaded our latest draft PDF, and re-run all ablation experiments** using the final FERN configuration that appears in the nonstationary shocks results (same data cleaning, 3-epoch grace period, EMA-smoothed validation, reflection=8 by default, patch size=24).  The qualitative conclusions are unchanged but clearer:

&nbsp;

**removing the encoder** (and latent mean updates) makes the model essentially unusable on ETTh2 and Lorenz-63;

&nbsp;

the “only encoder” setting (**no repeated translation conditioning**) slightly harms  ETTh2 but helps ETTh1 and especially Lorenz-63, highlighting the role of data-dependent shifts for chaotic systems;

&nbsp;

**removing rotation or patching** consistently degrades performance, confirming that local geometric alignment and patch-wise conditioning are both necessary;

&nbsp;

**increasing the number of reflections improves MSE/SWD/EPT up to 24 reflections**, at modest runtime cost.

&nbsp;

For transparency, we also now report wall-clock training times for each ablation configuration. No baselines were retrained; only FERN variants were updated to match the final architecture.

---

### Meta-Review · Area_Chair_Gbvi · 2026-01-07

**Summary:**

The paper presents a forecasting model that can better handle chaotic dynamics. The model is geometry-aware and learns the future geometry rather than a single trajectory through approximate locally linear ellipsoids. The authors also introduce new evaluation protocols to better test performance on chaotic systems. Their results show that FERN maintains the geometric reconstruction with competitive performance on standard benchmarks.

The reviewers all agree on the novelty of the paper, the strong results, and the critical stress-testing of existing approaches on chaotic systems and their proposed novel forecasting solution. However, all reviewers flagged a major concern on the presentation of the paper with several unhelpful mathematical concepts to motivate their approach and a dense presentation of their methodology that makes it difficult for the standard machine learning community reader to parse through.

The authors acknowledge this and re-write 85% of the paper. This is a significant effort on the authors' side. However, given the 85% rewrite, it is difficult to assess standalone whether the paper achieves the intended clarity. The paper still reads a bit dense and it's also uncertain whether the substantial rewrite has been sufficiently evaluated in its current form.

**Reviewer Concerns:**

There were important concerns raised on the demonstration of the results through helpful reconstruction plots and other diagnostics. The authors have addressed these in my opinion. There were several clarification questions on the concepts that the authors have attempted to answer and also incorporate into their re-write. However, there was not a general consensus yet. It is possible that the rewrite and more discussion would have persuaded reviewers to all come to a borderline accept.

**Reviewer Scores:**

The rebuttal has several new experiments and a significant rewrite. Given the general consensus on the positive results as well as the novel contribution but severely low scores (three 2s and one 6), it would require careful re-evaluation by the reviewers who expressed the strongest reservations. It is possible that the paper could have moved toward a borderline accept.

---

### Decision · Program_Chairs · 2026-01-26

Reject